# DATA-ADAPTIVE DIFFERENTIALLY PRIVATE PROMPT SYNTHESIS FOR IN-CONTEXT LEARNING

**Fengyu Gao**[*]
University of Virginia
wan6jj@virginia.edu

**Ruida Zhou**[*]
University of California Los Angeles
ruida@g.ucla.edu

**Tianhao Wang**
University of Virginia
tianhao@virginia.edu

**Cong Shen**
University of Virginia
cong@virginia.edu

**Jing Yang**
University of Virginia
yangjing@virginia.edu

## ABSTRACT

Large Language Models (LLMs) rely on the contextual information embedded in examples/demonstrations to perform in-context learning (ICL). To mitigate the risk of LLMs potentially leaking private information contained in examples in the prompt, we introduce a novel data-adaptive differentially private algorithm called **AdaDPSyn** to generate synthetic examples from the private dataset and then use these synthetic examples to perform ICL. The objective of AdaDPSyn is to adaptively adjust the noise level in the data synthesis mechanism according to the inherent statistical properties of the data, thereby preserving high ICL accuracy while maintaining formal differential privacy guarantees. A key innovation in AdaDPSyn is the *Precision-Focused Iterative Radius Reduction* technique, which dynamically refines the aggregation radius - the scope of data grouping for noise addition - based on patterns observed in data clustering, thereby minimizing the amount of additive noise. We conduct extensive experiments on standard benchmarks and compare AdaDPSyn with DP few-shot generation algorithm (Tang et al., 2023). The experiments demonstrate that AdaDPSyn not only outperforms DP few-shot generation, but also maintains high accuracy levels close to those of non-private baselines, providing an effective solution for ICL with privacy protection.

## 1 INTRODUCTION

In-context learning (ICL) (Brown et al., 2020; Min et al., 2022) enables large language models (LLMs) (OpenAI, 2023) to adapt to domain-specific information without modifying the pre-trained model. This adaptation is achieved by conditioning the model on a *prompt*, which contains an instruction and a series of task-specific question-answer pairs (called *demonstrations* or *examples*). Using this prompt, LLMs can then generate responses that are tailored to the corresponding task. The ease of usage and cost-benefit of ICL have motivated the adoption of LLMs in several applications, such as machine translation (Sia and Duh, 2023), code generation (Pourreza and Rafiei, 2023) and customer service (Lee et al., 2022).

However, privacy is a significant concern when incorporating users' data into prompts, especially in areas like healthcare and finance (Wang et al., 2023; Priyanshu et al., 2023; Duan et al., 2023), as this introduces risks of exposing sensitive personal information. To address this concern, existing works (Wu et al., 2023; Tang et al., 2023; Duan et al., 2023; Carey et al., 2024; Hong et al., 2023) have explored approaches to mitigate these risks using the rigorous privacy safeguards provided by differential privacy (DP) (Dwork et al., 2006b). Notably, Tang et al. (2023) proposed to generate synthetic few-shot demonstrations from the private dataset for use in ICL inference and guarantee DP with respect to examples in the private dataset. This method uses the classic DP sample-and-aggregate framework (Nissim et al., 2007; Papernot et al., 2016; 2018), partitioning subsampled private data into disjoint subsets, each of which is used in a prompt for an LLM to generate tokens. These tokens

---

[*]co-first author.

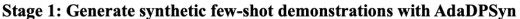

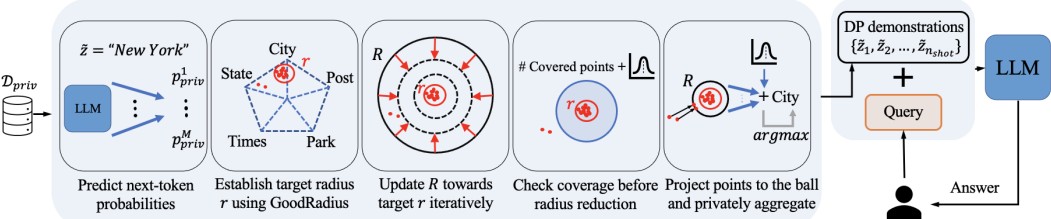

Figure 1: Two-Stage framework for privacy-preserving ICL. In Stage 1, DP-protected demonstrations are generated with AdaDPSyn, where we illustrate the process of generating the next token "City" following "New York". AdaDPSyn uses a novel clustering approach to dynamically aggregate next-token probabilities. The process iterates until $n_{\text{shot}}$ demonstrations are generated. In Stage 2, the generated DP-protected demonstrations are combined with a user query to construct the prompt. The constructed prompt is then sent to an LLM and the answer is returned to the user.

are then privately aggregated, adhering to DP guarantees, to create synthetic demonstrations. This framework allows the synthetic demonstrations to be used for an infinite number of queries without incurring any additional privacy costs.

The main design challenge in Tang et al. (2023) is aggregating the responses from an LLM privately. Tang et al. (2023) used a *data-independent* way of adding the same amount of Gaussian noise during private aggregation. However, since each token generated by the LLM can contain varying levels of private information, applying noise homogeneously across an entire sentence or document may not achieve the optimal privacy-utility tradeoff in the ICL framework. Therefore, it is more strategic to design DP mechanisms that adapt the noise level in a *data-adaptive* manner. Intuitively, by adaptively adjusting the additive noise level based on the properties of responses for individual tokens, we avoid adding unnecessarily large noise to tokens with high agreement or predictability, which indicates low risk, thereby achieving improved utility for ICL. The main question we aim to answer is:

*Can we design a data-adaptive differentially private prompt synthesis algorithm to protect the private information contained in the prompt data and effectively perform ICL?*

In this work, we provide an affirmative answer to this question. Our main contributions are summarized as follows:

- We introduce a new algorithm AdaDPSyn in Algorithm 1 to generate synthetic few-shot examples for use in ICL prompts (Figure 1). Similar to Tang et al. (2023), AdaDPSyn begins by obtaining next-token generation probability vectors from an LLM using multiple subsets of private dataset, which will then be aggregated to generate the next token differentially privately. The essential novelty of AdaDPSyn lies in our proposed *Precision-Focused Iterative Radius Reduction* technique, which adaptively adds noise level during aggregation by exploiting the cluster structure of the predicted distributions of the next token. Besides, we believe that our data-adaptive technique may be applicable in various problems beyond prompt synthesis, thus is of independent interest.

- We conduct a rigorous privacy analysis of AdaDPSyn and confirm it follows $(\varepsilon, \delta)$-DP. In our analysis, we use the Rényi Differential Privacy (RDP) (Mironov, 2017) framework to track and tightly quantify privacy costs from the composition of AdaDPSyn's components across multiple iterations. By applying RDP composition rules and subsampling amplification, we establish theoretical privacy guarantees for AdaDPSyn.

- We empirically evaluate the performance of AdaDPSyn against the DP few-shot generation algorithm proposed by Tang et al. (2023) on standard benchmarks. We demonstrate that our method outperforms this baseline across a plethora of privacy settings. For example, in the AGNews classification task under a strict privacy level of $\varepsilon = 1$, our method achieves an accuracy of $65.42\%$, higher than $60.74\%$ by DP few-shot generation. In the information extraction task MIT-G at $\varepsilon = 1$, our method reaches $37.59\%$ accuracy, compared to $31.15\%$ by DP few-shot generation. Additionally, we show that our AdaDPSyn algorithm can closely match the performance of non-private baselines even in stringent privacy settings. For instance, at a strict privacy level of $\varepsilon = 1$, our algorithm attains an accuracy of $65.42\%$ on the AGNews task, nearly reaching the non-private baseline's accuracy of $65.92\%$. Similarly, in the DBPedia task, our method shows minimal performance drop-off compared to the non-private scenarios. These results not only underscore the effectiveness of our AdaDPSyn algorithm compared with data-independent methods, but also demonstrate its competitiveness with non-private solutions.

## 2 RELATED WORK

**Differentially Private In-Context Learning and Fine-tuning of LLMs.** Duan et al. (2023) first study privacy leakage through membership inference attacks (MIAs) and suggest mitigating MIA risks using DP ensembling (Papernot et al., 2018) on various model versions. Recent studies (Wu et al., 2023; Tang et al., 2023; Duan et al., 2023; Carey et al., 2024) have developed methods based on DP ensembling (Papernot et al., 2018) to preserve privacy of the underlying prompt data. Notably, Tang et al. (2023) develop an algorithm that generates differentially private synthetic few-shot demonstrations from a private dataset for ICL inference, which serves as the baseline of this work and will be discussed in greater detail in subsequent sections. Hong et al. (2023) propose the DP-OPT algorithm that generates private and transferable instructions within prompts on the client side to be used with cloud-hosted LLMs. Carey et al. (2024) focus on protecting tabular data used for ICL. We note that all of those works (Tang et al., 2023; Carey et al., 2024; Hong et al., 2023) design private aggregation methods in a *data-independent* way, i.e., adding the same amount of noise during private aggregation.

Recently, Wu et al. (2023) and Duan et al. (2023) study differentially private ICL with *data-dependent* privacy analysis. Wu et al. (2023) propose DP inference by privately aggregating results from multiple queries using disjoint sampled demonstrations. They use the classic propose-test-release (PTR) paradigm (Zhu and Wang, 2022) to select the top-$K$ tokens from LLM outputs. If the vote count difference between the $K$-th and $(K+1)$-th highest tokens exceeds two, the tokens can be released without further privatization. However, ICL tasks can involve an unlimited number of queries, and *Wu et al. (2023) consume the privacy budget per query, limiting the number of queries that can be answered within a given budget.*

Duan et al. (2023) assume *the existence of an unlabeled public dataset*, which contrasts with our approach that requires no public data. In their method, an ensemble of teacher models uses private majority voting to label this public dataset. The labeled data then helps construct private prompts. Following Papernot et al. (2018), they focus on scenarios where a high agreement among teachers can reduce privacy costs below data-independent bounds. When teacher agreement is low, they discard the public data. In contrast, our method is better suited for scenarios where assuming the availability of unlabeled public data with a distribution similar to private data is unrealistic, such as in healthcare or industrial applications. Moreover, Duan et al. (2023) mainly focus on text classification tasks with a restricted label space, while our method applies to a broader range of tasks.

On the other hand, several studies have focused on private fine-tuning of LLMs (Yu et al., 2021; Li et al., 2021; Chen et al., 2024; Tang et al., 2024; Zhang et al., 2023; He et al., 2022). Besides, Majmudar et al. (2022) introduce DP at the decoding stage of a pre-trained LLM.

**Adaptive Differential Privacy.** Adaptive differential privacy primarily focuses on privately calibrating noise to the local sensitivity, which is a dataset-dependent function and is much smaller than the global sensitivity. Two representative solutions include the smooth sensitivity framework (Nissim et al., 2007) and the propose-test-release (PTR) framework (Dwork and Lei, 2009). The main idea of the smooth sensitivity framework is to add noise calibrated to the smooth sensitivity, an upper bound on the local sensitivity which changes slowly between neighboring datasets (Bun and Steinke, 2019; Gonem and Gilad-Bachrach, 2018; Fletcher and Islam, 2017; Zafarani and Clifton, 2020; Hamman et al., 2023; Sun et al., 2020). The PTR framework involves proposing bounds of the local sensitivity and testing its validity. If the test is passed, the noise is calibrated according to the proposed bound (Thakurta and Smith, 2013; Liu et al., 2022; Redberg et al., 2023; Wang et al., 2022; Liu et al., 2022).

In this work, we release a confidence bound of the local sensitivity in a differentially private manner, and calibrate noise accordingly, similar to the idea in prior studies (Blocki et al., 2012; Kasiviswanathan et al., 2013; Wang, 2018; Decarolis et al., 2020). In this body of research, Blocki et al. (2012) and Kasiviswanathan et al. (2013) focus on network data. Wang (2018) revisits the DP linear regression problem. Decarolis et al. (2020) study the Latent Dirichlet Allocation (LDA) model. Our focus is on the ICL domain, where we develop a novel clustering method and Precision-Focused Iterative Radius Reduction technique to address domain-specific challenges.

**Differentially Private Synthetic Text Generation.** Our work falls within the broader scope of DP synthetic text generation, which typically requires generating large volumes of synthetic data. There is a line of work on DP synthetic text generation through private fine-tuning (Yue et al.,

2022; Mattern et al., 2022; Mireshghallah et al., 2022; Carranza et al., 2023; Yu et al., 2024; Wu et al., 2024). SeqPATE (Tian et al., 2022) and Submix (Ginart et al., 2022) are methods based on PATE (Papernot et al., 2016; 2018) for text generation, which utilizes an ensemble of teacher models trained on private data subsets to fine-tune a student model. Another direction (Feyisetan et al., 2020; Xu et al., 2020; Du et al., 2023; Utpala et al., 2023) focuses on sanitizing user texts locally before server submission based on local differential privacy (Chatzikokolakis et al., 2013). While this approach typically reduces text utility, recent studies (Chen et al., 2022; Arnold et al., 2023b;a; Mireshghallah et al., 2022) propose to enhance utility by incorporating inherent properties of language such as semantic similarity (Chen et al., 2022), context (Arnold et al., 2023b) and syntax (Arnold et al., 2023a; Mireshghallah et al., 2022).

## 3 Problem Definition, Threat Model, and Notations

**In-context learning.** Given a query $x$, a candidate answer set (label set) $\mathcal{Y}$ and a pre-trained large language model LLM such as GPT (Radford et al., 2018; OpenAI, 2023), ICL aims to predict a label $y \in \mathcal{Y}$ of query $x$, using LLM conditioned on $n_{\text{shot}}$ demonstrations $\mathcal{C} = \left\{ (x'_1, y'_1), (x'_2, y'_2), \ldots, (x'_{n_{\text{shot}}}, y'_{n_{\text{shot}}}) \right\}$, where $(x'_i, y'_i)$ is an input-label pair.

In this paper, we consider both classification and information extraction tasks. For classification tasks with a restricted label space $\mathcal{Y} = \{y_1, \ldots, y_m\}$, we first compute the probability of each label $y_i \in \mathcal{Y}$ using LLM as $\mathbb{P}_{\text{LLM}}(y_i \mid \mathcal{C}, x)$. Then we predict the final label $y$ by selecting the label with the highest probability from the label set $\mathcal{Y}$, i.e., $y = \arg\max_{y_i \in \mathcal{Y}} \mathbb{P}_{\text{LLM}}(y_i \mid \mathcal{C}, x)$. For information extraction tasks, the label space $\mathcal{Y}$ is unrestricted, and can include any sequence of tokens from the LLM's vocabulary. In these tasks, we predict $y$ by having LLM generate a sequence of tokens until it produces a special end-of-sequence (EOS) token, marking the completion of the output.

**Differentially private ICL.** Our objective is to protect the privacy of demonstrations $\mathcal{C} \subseteq \mathcal{D}_{\text{priv}}$ in the prompt against an adversary that aims to access or infer private information about the demonstrations. Since ICL tasks may encounter an unlimited number of queries, to protect the privacy of the dataset $\mathcal{D}_{\text{priv}}$, we adopt a differentially private data synthesis approach to generate synthetic few-shot examples $\mathcal{C} = \{\widetilde{z}_1, \cdots, \widetilde{z}_{n_{\text{shots}}}\}$, where $\widetilde{z}_i = (\widetilde{x}_i, \widetilde{y}_i)$, from the underlying distribution of $\mathcal{D}_{\text{priv}}$ while satisfying $(\varepsilon, \delta)$-differential privacy on the private dataset $\mathcal{D}_{\text{priv}}$. For each ICL task, we generate task-specific synthetic examples $\mathcal{C}$ with the help of an LLM and a private dataset $\mathcal{D}_{\text{priv}}$. Then ICL can be performed using demonstrations from $\mathcal{C}$ in a prompt tailored to each task. By default, we incorporate all demonstrations from $\mathcal{C}$ in a prompt for each task.

We formally define $(\varepsilon, \delta)$-differential privacy $((\varepsilon, \delta)$-DP) as follows.

**Definition 1** $((\varepsilon, \delta)$-Differential Privacy (Dwork et al., 2006a)). *A randomized algorithm $A$ is $(\varepsilon, \delta)$-differentially private if for any two neighboring inputs $\mathcal{D}$ and $\mathcal{D}'$ that differ by a single entry and any set $\mathcal{S}$ of possible outputs: $\mathbb{P}[A(\mathcal{D}) \in \mathcal{S}] \leq e^{\varepsilon} \mathbb{P}[A(\mathcal{D}') \in \mathcal{S}] + \delta$.*

## 4 Proposed Method

We use a two-stage framework to perform differentially private ICL, as illustrated in Figure 1. Generally speaking, in **Stage 1**, we generate DP synthetic few-shot demonstrations $\mathcal{C} = \{\widetilde{z}_1, \cdots, \widetilde{z}_{n_{\text{shots}}}\}$ based on the private dataset $\mathcal{D}_{\text{priv}}$; and in **Stage 2**, we use these synthetic demonstrations for ICL.

The main challenge of this framework lies in the DP synthetic data generation (Stage 1). To maintain DP, a PATE-like framework is utilized, similar to Tang et al. (2023). Take the generation of a synthetic demonstration $\tilde{z} = (\tilde{x}, \tilde{y}) \in \mathcal{C}$ as an example. First, label $\tilde{y} \in \mathcal{Y}$ is randomly selected independently of private dataset $\mathcal{D}_{\text{priv}}$, and its corresponding demonstration $\tilde{x}$ is generated token by token, starting from an empty string $\tilde{x} = ``$. The next token for $\tilde{x}$ is generated by aggregating probability vectors $p_{\text{priv}}^1, \ldots, p_{\text{priv}}^M$, each of which is calculated by an LLM taking some prompt as input. Each prompt is a concatenation of a subset of examples (in a reverse order $(y_i, x_i)$) from the private dataset $\mathcal{D}_{\text{priv}} = \{(x_i, y_i)\}$, the synthetic label $\tilde{y}$ and the currently generated string $\tilde{x}$. Due to the dependence of the probability vectors on private dataset $\mathcal{D}_{\text{priv}}$, DP aggregation is required, and a classic way to achieve this is by adding noise (e.g., Gaussian mechanism) as in Tang et al. (2023).

Our approach is motivated by a hypothesis that the next-token predictions by different examples approximately reach a consensus. Once this hypothesis holds, the key innovation of our approach is to leverage this feature and reduce noise levels during aggregation in a data-adaptive manner. We verify the hypothesis quantitatively and tackle the key challenge in the following two sections, respectively.

### 4.1 CONSENSUS BEHAVIOR MEASURED BY CLUSTERING

We quantitatively measure the degree of consensus among the next-token generation probability vectors $p_{\text{priv}}^1, \ldots, p_{\text{priv}}^M$. Specifically, we approach this by visualizing consensus through the geometric concept of enclosing these vectors within a minimal enclosing ball, defined as an $(r, c, \rho)$-ball, where $r$ is the radius, $c$ is the center and $\rho \in (0, 1]$ represents the required portion of vectors covered in the ball. For a given coverage requirement $\rho$, a smaller $r$ indicates a tighter grouping, reflecting a high consensus among the model outputs. We hypothesize that *these next-token generation probability vectors are typically highly clustered in a ball with a small radius $r$*.

To verify the hypothesis, we conduct experiments using the DP few-shot generation algorithm (Tang et al., 2023), and report the radius $r$ of the minimal ball that encloses at least $80\%$ ($\rho = 0.8$) of the next-token probability vectors generated by using different private examples in Table 1. For determining a small radius $r$ that contains at least $80\%$ of the input points, we adopt the GoodRadius algorithm in Nissim et al. (2016). We conduct experiments on three classification tasks: AGNews (Zhang et al., 2015), DBPedia (Zhang et al., 2015) and TREC (Voorhees and Tice, 2000), as well as two information extraction tasks, MIT-G and MIT-D (Liu et al., 2012), using the Llama-2-7b-hf model (Touvron et al., 2023)[1]. Details of the DP few-shot generation algorithm parameters can be found in Table 12.

Table 1: The mean and standard deviation of $r$ over 5 runs with different random seeds.

| Task | radius $r$ |
|---|---|
| AGNews | $0.06_{0.02}$ |
| DBPedia | $0.05_{0.01}$ |
| TREC | $0.12_{0.01}$ |
| MIT-G | $0.15_{0.01}$ |
| MIT-D | $0.14_{0.01}$ |

As shown in Table 1, the measured radius $r$, ranging from 0.05 to 0.15, is much smaller than $\sqrt{2}/2$, the radius of the minimal ball that contains the probability simplex. These results confirm a high degree of clustering among the model outputs. Intuitively, if we can access such a ball of small radius $r$ differentially privately, we can perform a differentially private projected mean aggregation that projects the probability vectors to the ball and calculates the projected mean with additive Gaussian noise. The noise level is determined by the sensitivity $2r$. Note that in Tang et al. (2023), without leveraging the clustering behavior of the data, a much higher noise level corresponding to sensitivity $\sqrt{2}$ is employed. Comparing these two cases, the sensitivity $2r$ is approximately $5\times$ to $14\times$ smaller than $\sqrt{2}$, thus leading to more accurate aggregation due to lower noises. The key challenge is then how to differentially privately access a small ball that contains a majority number of probability vectors and perform the differentially private aggregation correspondingly. To tackle this challenge, we propose a novel Precision-Focused Iterative Radius Reduction technique.

### 4.2 PRECISION-FOCUSED ITERATIVE RADIUS REDUCTION

Building on the observation that next-token generation probability vectors from the LLM are highly clustered, we utilize an intuitive approach to adaptively adjust the noise level during aggregation: we first locate the minimal ball covering most vectors under DP and then project all points to this ball in $\ell_2$-norm to bound the influence of each point, thereby minimizing noise addition. This method, which adaptively leverages the inherent structure of the probability vectors, is essential to the success of our proposed method.

The task of finding a minimal enclosing ball under DP is known as the DP 1-clustering problem in the literature (Nissim et al., 2016). However, existing methods (Nissim et al., 2016; Nissim and Stemmer, 2018; Ghazi et al., 2020) are hard to implement and computationally expensive.

To this end, we propose a novel technique called **Precision-Focused Iterative Radius Reduction**. We start with a large radius $R$ and judiciously reduce it towards a small target radius $r$ through iterative refinement. The target radius $r$ is obtained under DP constraints using a tractable subroutine

---

[1]https://huggingface.co/meta-llama/Llama-2-7b-hf.

GoodRadius from DP 1-clustering (Nissim et al., 2016), which provides a threshold such that the designed radius should never go below. The reduction of radius $R$ is carefully monitored by coverage checks, which ensure the majority of the probability vectors are covered by a ball with that radius. The center of the ball is efficiently obtained by private projected mean estimation. This dynamic adjustment of the ball ensures it is minimized effectively according to the distribution of the points along the adjustments. The amount of noise required for private projected mean estimation can be reduced, thus improving the quality of the synthetic demonstrations and the performance of ICL. We elaborate this Precision-Focused Iterative Radius Reduction technique in Section 4.3.

## 4.3 ADADPSYN ALGORITHM

We now introduce the proposed AdaDPSyn algorithm (Algorithm 1), which generates DP synthetic few-shot examples from the private dataset $\mathcal{D}_{\text{priv}}$. For a given label $\tilde{y}$ randomly selected from the label set without replacement, AdaDPSyn sequentially generates one token at a time, starting from an empty list. Each token generation begins with the Next Token Generation subroutine (Line 4). Roughly speaking, the Next Token Generation process (Tang et al., 2023) involves sampling $MN$ examples with label $\tilde{y}$ from the private dataset $\mathcal{D}_{\text{priv}}$, dividing them into $M$ disjoint subsets, and then using these subsets as demonstrations to predict the next token's distribution $p_{\text{priv}}^1, \ldots, p_{\text{priv}}^M$ using an LLM. The vocabulary is restricted to $\mathcal{S}$ containing the top-$K$ most probable tokens, which are determined from next-token probabilities generated only from instructions without using any private data. Details of Next Token Generation can be found in Appendix A.1. Then we privately aggregate $p_{\text{priv}}^1, \ldots, p_{\text{priv}}^M$ using our *Precision-Focused Iterative Radius Reduction* technique (Lines 5-21), which consists of four critical steps:

**Step 1: Establish Target Radius Differentially Privately.** The process begins by setting a target radius $r$ using the GoodRadius subroutine (Nissim et al., 2016) in Line 5. GoodRadius is a subroutine used in DP 1-clustering methods (Nissim et al., 2016; Nissim and Stemmer, 2018), which identifies a radius that can encompass at least a fraction $\rho$ of the data points under $(\alpha, \tau_0)$-RDP constraints. In this work, we set $\rho = 80\%$. Details are provided in Appendix A.2.

**Step 2: Projected Mean Estimation Using Gaussian DP Mechanism.** For a probability simplex in $\mathcal{R}^d$, its radius[2] is $\frac{\sqrt{2}}{2}$. Thus, we set $R \leftarrow \sqrt{2}/2$ (Line 6) initially. We initialize projected points $\tilde{p}_{\text{priv}}^i \leftarrow p_{\text{priv}}^i, i = 1, \ldots, M$ (Line 7), and estimate mean of $\{\tilde{p}_{\text{priv}}^1, \ldots, \tilde{p}_{\text{priv}}^M\}$ using Gaussian DP mechanism (Line 8), i.e., $\tilde{p}_{\text{priv}} \leftarrow \frac{1}{M}\left(\sum_{i=1}^M \tilde{p}_{\text{priv}}^i + \mathcal{N}(0, 4R^2\sigma_1^2 I)\right)$, where $\sigma_1$ is the noise multiplier. Then we map $\tilde{p}_{\text{priv}}$ back onto the probability simplex, i.e., $\tilde{p}_{\text{priv}} \leftarrow \frac{\max(\tilde{p}_{\text{priv}}, 0)}{\|\max(\tilde{p}_{\text{priv}}, 0)\|_1}$ (Line 9).

Next, we reduce $R$ towards the target radius $r$ iteratively, ensuring that it covers a sufficient amount of probability vectors while satisfying DP (Lines 11-21). During the iterative process, which takes at most $\hat{T}$ iterations, each step starts with a radius coverage check detailed in the following paragraph.

**Step 3: Radius Coverage Check Using Gaussian DP Mechanism.** We use Gaussian DP mechanism to check whether a sufficient number of the original probability vectors $p_{\text{priv}}^1, \ldots, p_{\text{priv}}^M$ remain within a small radius from $\tilde{p}_{\text{priv}}$ (Line 12). Specifically, the check evaluates whether the $\ell_2$-ball with center $\tilde{p}_{\text{priv}}$ and radius $r + \frac{2\lambda R \sigma_1 \sqrt{K}}{M}$ can cover at least $\mu M$ of the points $p_{\text{priv}}^1, \ldots, p_{\text{priv}}^M$, where $\lambda$ is a hyperparameter, $\sigma_1$ scales the Gaussian noise introduced to $\tilde{p}_{\text{priv}}$ in Line 8 or Line 20, $K$ is the dimensionality of the probability vectors, and $\mu \in (0, 1]$. We set $\mu = 0.55$ in this work. The term $\frac{2\lambda R \sigma_1 \sqrt{K}}{M}$ represents an additional margin on the radius to accommodate the spread introduced by Gaussian noise added to $\tilde{p}_{\text{priv}}$ in Line 8 or Line 20, ensuring data coverage despite the distortion caused by the noise.

**Step 4: Update Radius towards Target Radius.** If the radius coverage check in Line 12 reveals that fewer than $\mu M$ vectors are within the defined radius, the algorithm will stop reducing the current radius $R$. Otherwise, the algorithm proceeds to adjust $R$ downward (Line 15-18), i.e., $R \leftarrow r +$

---

[2]For a bounded region $\mathcal{R} \subset \mathbb{R}^d$, we define its diameter as $\text{diam}(\mathcal{R}) = \sup_{x,y \in \mathcal{R}} \|x - y\|$, and its radius as half of the diameter, i.e., $\text{Rad}(\mathcal{R}) = \frac{1}{2}\text{diam}(\mathcal{R})$. The definition of diameter coincides with the definition of sensitivity for the aggregation (summation) of a utility function, thus the radius can be interpreted as half of the sensitivity factor.

$\frac{2\lambda R \sigma_1 \sqrt{K}}{M}$. Then we project next-token probability vectors $\{p_{\text{priv}}^1, \ldots, p_{\text{priv}}^M\}$ to a $\ell_2$-ball defined by radius $R$ and centered at $\tilde{p}_{\text{priv}}$ (Line 19), i.e., $\tilde{p}_{\text{priv}}^i = \tilde{p}_{\text{priv}} + (p_{\text{priv}}^i - \tilde{p}_{\text{priv}}) \big/ \max\left(1, \frac{\|p_{\text{priv}}^i - \tilde{p}_{\text{priv}}\|_2}{R}\right)$, $i = 1, \ldots, M$. This operation ensures that the $\ell_2$-sensitivity of the term $\sum_{i=1}^M \tilde{p}_{\text{priv}}^i$ is at most $2R$. Finally, we estimate the mean of the projected points $\{\tilde{p}_{\text{priv}}^1, \ldots, \tilde{p}_{\text{priv}}^M\}$ using Gaussian DP mechanism (Line 20), i.e., $\tilde{p}_{\text{priv}} \leftarrow \frac{1}{M}\left(\sum_{i=1}^M \tilde{p}_{\text{priv}}^i + \mathcal{N}(0, 4R^2\sigma_1^2 I)\right)$. As $R$ decreases, the amount of noise added to $\tilde{p}_{\text{priv}}$ is reduced accordingly. We then map $\tilde{p}_{\text{priv}}$ back onto the probability simplex, i.e., $\tilde{p}_{\text{priv}} \leftarrow \frac{\max(\tilde{p}_{\text{priv}}, 0)}{\|\max(\tilde{p}_{\text{priv}}, 0)\|_1}$ (Line 21).

The process of radius refinement concludes under any of the following conditions: (1) the maximum allowed number of iterations, $\hat{T}$, is reached; (2) further reduction in radius fails to cover the minimum required proportion of vectors (Line 12); (3) the current radius $R$ is smaller than $r + \frac{2\lambda_1 R \sigma_1 \sqrt{K}}{M}$ (Line 15). Once the radius refinement process concludes, we select the next token for the synthetic sequence as $\arg\max_{j \in \mathcal{S}} \tilde{p}_{\text{priv}}[j]$ where $\mathcal{S}$ is the vocabulary of size $K$, and this token is appended to what is generated so far. This continues until the sequence reaches the predefined maximum limit, $T_{\max}$, of tokens we would like to generate.

*Remark* 1 (Comparison with Tang et al. (2023)). The main difference between our method and that of Tang et al. (2023) lies in the innovative approach we take toward the private aggregation of the next-token generation probabilities $p_{\text{priv}}^1, \ldots, p_{\text{priv}}^M$. Instead of directly using the Gaussian DP mechanism for aggregation as in Tang et al. (2023), our approach introduces an advancement through the Precision-Focused Iterative Radius Reduction technique (Lines 5-21). This technique allows noise to be added in a data-adaptive manner, effectively adapting the noise level to the degree of agreement among the models. Empirically, we evaluate the performance of our algorithm against Tang et al. (2023) in Section 5.1 to demonstrate the effectiveness of our method.

### 4.4 PRIVACY ANALYSIS

**Theorem 1.** *Algorithm 1 is $(\varepsilon, \delta)$-differentially private.*

*Proof Overview.* We adopt the commonly used Rényi differential privacy (RDP) (Mironov, 2017) to track the privacy cost in our algorithm, as it allows us to tightly quantify the privacy guarantees from the composition of multiple mechanisms. Our proof mainly consists of three steps. First, we show that each iteration of the algorithm is $(\alpha, \tau)$-RDP. In each iteration, our algorithm consists of 3 components: one instance of $(\alpha, \tau_0)$-RDP algorithm GoodRadius (Line 5), at most $(\hat{T}+1)$ instances of $(\alpha, \tau_1)$-RDP projected mean estimation using Gaussian DP Mechanism (Line 8 and Line 20) and at most $\hat{T}$ instances of $(\alpha, \tau_2)$-RDP radius coverage check using Gaussian DP Mechanism (Line 12). We apply the composition property of RDP to ensure the privacy guarantee in each iteration, that is, we ensure that $\tau = \tau_0 + (\hat{T}+1)\tau_1 + \hat{T}\tau_2$. Next, since the Next Token Generation subroutine (Line 4) involves drawing $MN$ samples from the dataset $\mathcal{D}_{\text{priv}}$, there is a privacy amplification by subsampling at each iteration. Finally, we compose privacy loss across all the $T_{\max}$ iterations of our algorithm using composition theorems. We rely on the conversion lemma (Balle et al., 2020) to convert the RDP guarantee back to DP notions. We provide a full proof in Appendix B. $\qquad\square$

## 5 EXPERIMENTS

**Datasets.** We study text classification on three datasets: 4-way news classification AGNews (Zhang et al., 2015), 6-way question classification TREC (Voorhees and Tice, 2000), and 14-way topic classification DBPedia (Zhang et al., 2015). For information extraction, we study the MIT Movies trivia10k13 slot-filling dataset (Liu et al., 2012), which includes movie genre (MIT-G) and director name (MIT-D) as slots. Additional dataset details are available in Appendix C.2.

**Setups.** We use the synthetic few-shot demonstrations $\{\tilde{z}_1, \cdots, \tilde{z}_{n_{\text{shots}}}\}$ from Algorithm 1 as input demonstrations in ICL for the aforementioned downstream tasks. We fix $n_{\text{shots}} = 4$ by default, generating 4-shot demonstrations (randomly without replacement from the label set) for ICL. We use Llama-2-7b-hf model (Touvron et al., 2023) as our pre-trained LLM to generate synthetic

---

**Algorithm 1** AdaDPSyn

---

1: **Input:** Private dataset: $\mathcal{D}_{\text{priv}}$, label: $\tilde{y}$, maximum token count: $T_{\max}$, an LLM: $\text{LLM}(\cdot)$, noise multiplier: $\sigma_0, \sigma_1, \sigma_2$, number of disjoint subsets: $M$, samples per subset: $N$, reduced vocabulary size: $K$, hyperparameters: $\lambda$ and $\hat{T}$.

2: **Initialize:** Set $\tilde{x} \leftarrow []$.

3: **for** $t = 1$ to $T_{\max}$ **do**

4:      $p_{\text{priv}}^1, \ldots, p_{\text{priv}}^M \leftarrow \text{NextTokenGeneration}\left(\mathcal{D}_{\text{priv}}, \tilde{y}, \text{LLM}(\cdot), M, N, K, \tilde{x}\right)$

5:      $r \leftarrow \text{GoodRadius}(\{p_{\text{priv}}^1, \ldots, p_{\text{priv}}^M\}, \rho M, \sigma_0)$        $\triangleright$ `Establish Target Radius`

6:      $R \leftarrow \sqrt{2}/2$

7:      $\tilde{p}_{\text{priv}}^i \leftarrow p_{\text{priv}}^i, i = 1, \ldots, M$

8:      $\tilde{p}_{\text{priv}} \leftarrow \frac{1}{M}\left(\sum_{i=1}^M \tilde{p}_{\text{priv}}^i + \mathcal{N}(0, 4R^2\sigma_1^2 I)\right)$

9:      $\tilde{p}_{\text{priv}} \leftarrow \frac{\max(\tilde{p}_{\text{priv}}, 0)}{\|\max(\tilde{p}_{\text{priv}}, 0)\|_1}$

10:      **for** $j = 1$ to $\hat{T}$ **do**

11:          $t \leftarrow$ the count of $p_{\text{priv}}^1, \ldots, p_{\text{priv}}^M$ within a $\ell_2$-ball centered at $\tilde{p}_{\text{priv}}$ with radius $r + \frac{2\lambda R\sigma_1\sqrt{K}}{M}$

12:          **if** $t + \mathcal{N}(0, \sigma_2^2) < \mu M$ **then**

13:              **break**                         $\triangleright$ `DP Radius Coverage Check`

14:          **end if**

15:          **if** $R < r + \frac{2\lambda R\sigma_1\sqrt{K}}{M}$ **then**

16:              **break**

17:          **end if**

18:          $R \leftarrow r + \frac{2\lambda R\sigma_1\sqrt{K}}{M}$          $\triangleright$ `Update Radius towards Target Radius`

19:          $\tilde{p}_{\text{priv}}^i \leftarrow \tilde{p}_{\text{priv}} + (p_{\text{priv}}^i - \tilde{p}_{\text{priv}})\Big/\max\left(1, \frac{\|p_{\text{priv}}^i - \tilde{p}_{\text{priv}}\|_2}{R}\right), i = 1, \ldots, M$

20:          $\tilde{p}_{\text{priv}} \leftarrow \frac{1}{M}\left(\sum_{i=1}^M \tilde{p}_{\text{priv}}^i + \mathcal{N}(0, 4R^2\sigma_1^2 I)\right)$    $\triangleright$ `DP Projected Mean Estimation`

21:          $\tilde{p}_{\text{priv}} \leftarrow \frac{\max(\tilde{p}_{\text{priv}}, 0)}{\|\max(\tilde{p}_{\text{priv}}, 0)\|_1}$

22:      **end for**

23:      $w \leftarrow \arg\max_{j \in S} \tilde{p}_{\text{priv}}[j]$

24:      $\tilde{x} \leftarrow \tilde{x} + [w]$

25: **end for**

26: **return** $\tilde{x}$.

---

demonstrations[3]. We use the same prompt format during ICL following Tang et al. (2023) (see Appendix C.11). For ICL downstream tasks, we use Llama-2-7b-hf for AGNews, DBPedia, MIT-G, and MIT-D, and GPT-4o mini for TREC[4]. For DP algorithms, we follow the common practice to set the privacy budget $\delta = 1/|\mathcal{D}_{\text{priv}}|$ (Tang et al., 2023; Hong et al., 2023).

**Baselines.** We compare our AdaDPSyn algorithm with DP few-shot generation in Tang et al. (2023), which uses a data-independent approach of adding the same level of Gaussian noise during aggregation. Hypermarameters in DP few-shot generation are detailed in Table 12, including the number of subsets $M$, the number of data samples per subset $N$, the number of tokens $T_{\max}$, and the reduced vocabulary size $K$. To ensure a fair comparison, we select these parameters for DP few-shot generation based on the guidance in Tang et al. (2023), as detailed in Appendix C.5. We then use the same values of $M$, $N$, $T_{\max}$ and $K$ for our AdaDPSyn without any additional hyperparameter tuning.

As a fully private baseline $\varepsilon = 0$, we follow Tang et al. (2023) to generate synthetic 4-shot demonstrations using purely instructions without any private data[5]. We consider two non-private baselines

---

[3] Next Token Generation (Tang et al., 2023) uses OpenAI's API logprobs parameters with a value of 100, which currently supports a maximum value of 5. To overcome this, we use the Llama-2-7b-hf model via vLLM platform (https://github.com/vllm-project/vllm), which offers a compatible API, as suggested in Tang et al. (2023).

[4] When using Llama-2-7b-hf for TREC, we observe high standard deviation in the results (around 10), likely due to the complexity of the task. GPT-4o mini reduces the standard deviation to around 5.

[5] Tang et al. (2023) demonstrate that in some applications, LLMs can generate relevant few-shot demonstrations using only instructions and perform competitively with DP few-shot generation algorithm.

Table 2: 4-shot ICL performance on the test set of downstream tasks with baselines. Results show the mean and standard deviation of accuracy over 5 runs. Our private solution with various privacy levels $\varepsilon = 1, 2, 4, 8$ uses AdaDPSyn to generate DP synthetic few-shot demonstrations for ICL, in comparison with the DP few-shot generation algorithm (Tang et al., 2023). $\varepsilon = 0$ represents a fully private solution. $\varepsilon = \infty$ serves as the non-private baseline.

| Dataset | Method | $\varepsilon = 0$ | $\varepsilon = 1$ | $\varepsilon = 2$ | $\varepsilon = 4$ | $\varepsilon = 8$ | $\varepsilon = \infty$ |
|---|---|---|---|---|---|---|---|
| AGNews | DP few-shot generation | $61.38_{7.10}$ | $60.74_{3.09}$ | $62.64_{1.82}$ | $62.70_{1.30}$ | $63.18_{1.26}$ | $65.92_{1.13}$ |
| | **AdaDPSyn** | | $\mathbf{65.42}_{1.08}$ | $\mathbf{65.52}_{0.81}$ | $\mathbf{65.84}_{1.03}$ | $\mathbf{65.92}_{1.02}$ | |
| DBPedia | DP few-shot generation | $63.92_{3.07}$ | $64.68_{1.42}$ | $64.78_{1.29}$ | $64.92_{1.99}$ | $65.26_{2.12}$ | $69.06_{1.40}$ |
| | **AdaDPSyn** | | $\mathbf{66.76}_{1.45}$ | $\mathbf{67.48}_{1.54}$ | $\mathbf{67.12}_{1.09}$ | $\mathbf{67.70}_{1.68}$ | |
| TREC | DP few-shot generation | $59.40_{2.62}$ | $62.48_{9.38}$ | $65.28_{5.67}$ | $66.12_{5.46}$ | $66.52_{5.57}$ | $72.32_{2.76}$ |
| | **AdaDPSyn** | | $\mathbf{69.32}_{7.19}$ | $\mathbf{69.80}_{4.57}$ | $\mathbf{71.48}_{2.93}$ | $\mathbf{72.08}_{2.24}$ | |
| MIT-G | DP few-shot generation | $13.62_{6.51}$ | $31.15_{4.04}$ | $34.00_{5.13}$ | $35.85_{3.63}$ | $36.10_{5.52}$ | $43.23_{7.00}$ |
| | **AdaDPSyn** | | $\mathbf{37.59}_{5.40}$ | $\mathbf{37.41}_{4.36}$ | $\mathbf{37.85}_{4.25}$ | $\mathbf{38.31}_{5.03}$ | |
| MIT-D | DP few-shot generation | $52.29_{9.43}$ | $72.87_{3.48}$ | $73.11_{1.86}$ | $74.60_{2.29}$ | $75.18_{3.74}$ | $79.33_{1.48}$ |
| | **AdaDPSyn** | | $\mathbf{73.35}_{2.13}$ | $\mathbf{74.41}_{2.55}$ | $\mathbf{75.08}_{1.30}$ | $\mathbf{75.28}_{1.93}$ | |

for $\varepsilon = \infty$: (i) DP few-shot generation algorithm operates without any added noise; (ii) 4-shot demonstrations randomly selected from the private dataset. We present the best of two accuracies between the two baselines. Specifically, (i) performs better for AGNews, TREC, and MIT-G, while (ii) performs better for DBPedia and MIT-D, as detailed in Table 5.

It is noteworthy that, on the AGNews task, the fully private baseline $\varepsilon = 0$ outperforms baseline DP few-shot generation with higher $\varepsilon$ values. This aligns with findings from Tang et al. (2023) (Table 1 in Tang et al. (2023)[6]), which highlights that LLMs are capable of generating meaningful demonstrations purely from instructions in some applications, performing well even without private data. Meanwhile, AdaDPSyn outperforms the $\varepsilon = 0$ baseline at $\varepsilon = 1, 2, 4, 8$, showing the advantage of our data-adaptive design.

## 5.1 MAIN RESULTS

We present our main results in Table 2. We provide the mean and standard deviation of the accuracy on the test data with ICL over 5 runs with different random seeds. In general, our results demonstrate that AdaDPSyn outperforms DP few-shot generation (Tang et al., 2023) across various privacy settings, while also closely approximating the performance of the non-private baseline.

We observe that, compared with DP few-shot generation, AdaDPSyn provides gains at privacy levels $\varepsilon = 1, 2, 4, 8$. For instance, in the AGNews news classification task at $\varepsilon = 1$, our method achieves an accuracy of 65.42% compared to 60.74% under DP few-shot generation. In the DBPedia topic classification, we reach 66.76%, surpassing the previous 64.68%. Moreover, in the information extraction tasks, our performance improvements are also evident with MIT-G achieving 37.59% over 31.15% at $\varepsilon = 1$. This highlights our method's efficiency in adapting noise addition to the degree of consensus among model outputs, improving accuracy for ICL.

Additionally, our method closely approaches the performance of the non-private baseline $\varepsilon = \infty$ on several tasks. Specifically, for news classification AGNews with a strict privacy level as small as $\varepsilon = 1$, our method achieves an accuracy of 65.42%, just slightly below the non-private baseline's accuracy of 65.92%. For topic classification DBPedia at $\varepsilon = 1$, we record an accuracy of 66.76%, near the baseline's 69.06%. In the question classification task TREC at $\varepsilon = 1$, our approach reaches 69.32%, closely following the baseline's 72.32%.

We conduct a hyperparameter search for AdaDPSyn on $\lambda$ and $\hat{T}$ in Appendix C.4. We observe that ICL accuracy remains stable across different $\lambda$ and $\hat{T}$ settings. For example, on AGNews, AdaDPSyn achieves accuracy between 64.28% and 65.92% for $\hat{T} \in \{1, 2\}$ and $\lambda \in \{0.15, 0.2, 0.25\}$, compared to 63.18% for DP few-shot generation when $\varepsilon = 8$. This highlights the stability of Algorithm 1. We provide the parameters used for our main results in Appendix C.5.

---

[6]In Tang et al. (2023), Table 1 shows that for AGNews using GPT-3 Babbage, the $\varepsilon = 0$ result outperforms DP few-shot generation at $\varepsilon = 1, 2$.

## 5.2 ABLATION STUDIES

**Varying number of shots.** Using the DBPedia dataset and setting $\varepsilon = 4$, we test the number of shots for ICL with $n_{\text{shots}} = 1, 2, 4, 8$. The results are presented in Table 3. $\varepsilon = 4$ (AdaDPSyn) presents the performance of our private solution and $\varepsilon = 4$ (DP few-shot generation) is based on the solution in Tang et al. (2023). We present two non-private performances: (i) DP few-shot generation algorithm operates without any added noise; (ii) few-shot demonstrations randomly selected from the private dataset.

We observe that AdaDPSyn consistently outperforms DP few-shot generation across all $n$-shot settings. Additionally, increasing the number of shots can improve the performance of AdaDPSyn. This suggests that AdaDPSyn can benefit from larger $n$-shot scenarios.

Table 3: ICL performance on the test set of the DBPedia dataset with various number of shots. $\varepsilon = 4$ (AdaDPSyn) uses AdaDPSyn to generate DP synthetic few-shot demonstrations for ICL. $\varepsilon = 4$ (DP few-shot generation) is based on the solution in (Tang et al., 2023). $\varepsilon = \infty$ (Algorithm 2 with $\sigma = 0$) represents DP few-shot generation algorithm operating without any added noise. $\varepsilon = \infty$ (rand) represents using demonstrations randomly selected from private dataset. Results show the mean and standard deviation of accuracy over 5 runs.

|  | $n_{\text{shots}} = 1$ | $n_{\text{shots}} = 2$ | $n_{\text{shots}} = 4$ | $n_{\text{shots}} = 8$ |
|---|---|---|---|---|
| $\varepsilon = 4$ (AdaDPSyn) | $65.94_{0.27}$ | $67.00_{1.00}$ | $67.12_{1.09}$ | $67.55_{1.27}$ |
| $\varepsilon = 4$ (DP few-shot generation) | $64.44_{1.55}$ | $65.06_{1.98}$ | $64.92_{1.99}$ | $66.32_{1.57}$ |
| $\varepsilon = \infty$ (Algorithm 2 with $\sigma = 0$) | $67.58_{0.77}$ | $67.82_{0.34}$ | $67.88_{1.75}$ | $68.16_{1.61}$ |
| $\varepsilon = \infty$ (rand) | $69.02_{2.21}$ | $69.38_{2.66}$ | $69.06_{1.40}$ | $69.68_{1.29}$ |

**Varying LLMs.** We evaluate ICL performance across different LLMs and present the results in Table 4. Specifically, we use the generated examples from Section 5.1 as demonstrations with GPT-3.5 Turbo and GPT-4o mini, available through OpenAI's service, for downstream ICL tasks. The experiments are conducted on the DBPedia dataset.

We observe that AdaDPSyn consistently outperforms DP few-shot generation and performs comparably to non-private baselines. Performance improves with both GPT-4o mini and GPT-3.5 Turbo for all models, with GPT-4o mini showing the highest overall results. Our method remains open to further improvements with more advanced LLMs.

Table 4: ICL performance on test set of DBPedia dataset with various LLMs. $\varepsilon = 4$ (AdaDPSyn) uses AdaDPSyn to generate DP synthetic few-shot demonstrations for ICL. $\varepsilon = 4$ (DP few-shot generation) is based on the solution in (Tang et al., 2023). $\varepsilon = \infty$ (Alg 2, $\sigma = 0$) represents DP few-shot generation algorithm operating without any added noise. $\varepsilon = \infty$ (rand) represents using demonstrations randomly selected from private dataset.

| Model | $\varepsilon = 4$ (DP few-shot generation) | $\varepsilon = 4$ (AdaDPSyn) | $\varepsilon = \infty$ (Alg 2, $\sigma = 0$) | $\varepsilon = \infty$ (rand) |
|---|---|---|---|---|
| LLama-2-7B-hf | $64.92_{1.99}$ | $67.12_{1.09}$ | $67.88_{1.75}$ | $69.06_{1.40}$ |
| GPT-3.5 Turbo | $89.94_{0.85}$ | $91.92_{1.46}$ | $92.16_{1.19}$ | $93.26_{0.45}$ |
| GPT-4o mini | $91.40_{1.09}$ | $93.08_{1.09}$ | $93.28_{1.24}$ | $94.32_{0.35}$ |

**Performance against membership inference attacks (MIAs).** Additionally, we conduct an empirical privacy evaluation in Appendix C.6 using membership inference attacks (MIA) (Shokri et al., 2017; Duan et al., 2023). Our results show that, in contrast to non-private ICL, which suffers from significant membership privacy leakage, our AdaDPSyn algorithm effectively reduces the MIA success rate to the level of random guessing.

## 6 CONCLUSION

In this work, we introduced the AdaDPSyn algorithm, a novel approach to ICL that incorporates DP to safeguard sensitive data used in LLM prompts. By leveraging a data-adaptive noise addition strategy through our Precision-Focused Iterative Radius Reduction technique, we effectively reduced noise levels without compromising DP guarantees, thus maintaining higher accuracy for ICL. Our empirical results demonstrate the superior performance of AdaDPSyn, which nearly matches the non-private baseline's performance on several benchmarks. One limitation of this work is the lack of theoretical guarantees for utility, as it is difficult to find an appropriate closed-form expression of ICL accuracy. We leave this exploration to future work.

## ACKNOWLEDGMENTS

The work of F. Gao and J. Yang was supported in part by the U.S. National Science Foundation under the grant ECCS-2133170. The work of C. Shen was supported in part by the U.S. National Science Foundation under the grants CNS-2002902, ECCS-2029978, ECCS-2143559, CPS-2313110, and ECCS-2332060. The work of T. Wang was supported in part by the U.S. National Science Foundation under the grant CNS-2220433. The work of R. Zhou was supported in part by the U.S. National Science Foundation under the grants 2139304, 2146838 and the Army Research Laboratory grant under Cooperative Agreement W911NF-17-2-0196.

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

# A   DP ALGORITHMS

In this section, we list all the differentially private mechanisms that are used in Algorithm 1 and provide their guarantees.

## A.1   DP FEW-SHOT GENERATION

The DP few-shot generation algorithm (Tang et al., 2023) is detailed in Algorithm 2. The Next Token Generation subroutine (Tang et al., 2023) is presented in Algorithm 3.

---

**Algorithm 2** DP few-shot generation (Tang et al., 2023)

---

1: **Input:** Private dataset: $\mathcal{D}_{\text{priv}}$, label: $\tilde{y}$, max number of tokens to generate: $T_{\max}$, a pre-trained LLM: $\text{LLM}(\cdot)$, number of disjoint subsets of private data: $M$, number of data samples in each subset: $N$, reduce vocab publicly with top-$K$, noise multiplier: $\sigma$.
2: **Algorithm used:** Algorithm 3 for generating next token probilities from LLM.
3: **Initialize:** Set $\tilde{x} \leftarrow []$.
4: **for** $t = 1$ to $T_{max}$ **do**
5:    $p_{\text{priv}}^1, \ldots, p_{\text{priv}}^M \leftarrow \text{NextTokenGeneration}\left(\mathcal{D}_{\text{priv}}, \tilde{y}, \text{LLM}(\cdot), M, N, K, \tilde{x}\right)$
6:    $p_{\text{priv}} \leftarrow \frac{1}{M}\left(\sum_{i=1}^M p_{\text{priv}}^i + \mathcal{N}(0, 2\sigma^2 I)\right)$
7:    $w \leftarrow \arg\max_{j \in S} p_{\text{priv}}[j]$
8:    $\tilde{x} \leftarrow \tilde{x} + [w]$
9: **end for**
10: **return** $\tilde{x}$

---

**Algorithm 3** Next Token Generation (Tang et al., 2023)

---

1: **Input:** Private dataset: $\mathcal{D}_{\text{priv}}$, label: $\tilde{y}$, a pre-trained LLM: $\text{LLM}(\cdot)$, number of disjoint subsets of private data: $M$, number of data samples in each subset: $N$, reduce vocab publicly with top-$K$, generated tokens $\tilde{x}$.
2: $\mathcal{D}'_{\text{priv}} \leftarrow$ randomly draw $MN$ samples from $\mathcal{D}_{\text{priv}}$ with label $\tilde{y}$
3: $p \leftarrow \text{LLM}(\cdot | PB(instruction, \tilde{y}, \tilde{x}))$ where $PB(\cdot)$ is defined in Appendix C.11
4: **for** $i = 1$ to $M$ **do**
5:    $\mathcal{D}_{\text{priv}}^{(i)} \leftarrow \mathcal{D}'_{\text{priv}}[(i-1)N : iN]$
6:    $p_{\text{priv}}^i \leftarrow \text{LLM}(\cdot | PB(instruction, \mathcal{D}_{\text{priv}}^{(i)}, \tilde{y}, \tilde{x}))$
7:    $S \leftarrow$ top-$K$ indices of $p$
8:    $p_{\text{priv}}^i[\mathcal{V} \setminus S] \leftarrow 0$ and re-scale $p_{\text{priv}}^i$ s.t. $\sum p_{\text{priv}}^i[S] = 1$
9: **end for**
10: **return** $p_{\text{priv}}^1, \ldots, p_{\text{priv}}^M$

---

## A.2   GOODRADIUS

Nissim et al. (2016) propose the GoodRadius algorithm (Algorithm 4) and provide its DP analysis. We adapt this with some modifications to provide an RDP guarantee. For readability, we include the RDP analysis here.

**Theorem 2** (Nissim et al. (2016)). *GoodRadius (Algorithm 4) is $(\alpha, \tau_0)$-RDP.*

*Proof.* From Nissim et al. (2016), the sensitivity of $L$ defined in Line 3 of Algorithm 4 is 2. Then in Line 4, we search for an $r$ such that $L(r) \geq t$ and $L(r/2) < t$ while maintaining differential privacy. This is achieved by using binary search with noisy estimates of $L$ for the comparisons. Specifically, we introduce a tolerance parameter $\theta$, which determines the precision of the binary search. The number of iterations for the binary search is at most $\lceil \log_2\left(\frac{\sqrt{2}}{2\theta}\right) \rceil$. In each iteration, we apply Gaussian mechanism to estimate $L(r)$ and $L(r/2)$, i.e., $\tilde{L}(r_{\text{mid}}) \leftarrow L(r_{\text{mid}}) + \mathcal{N}(0, 4\sigma_0^2)$ and $\tilde{L}(r_{\text{mid}}/2) \leftarrow L(r_{\text{mid}}/2) + \mathcal{N}(0, 4\sigma_0^2)$. By setting $\sigma_0 \leftarrow \sqrt{\alpha \lceil \log_2\left(\frac{\sqrt{2}}{2\theta}\right) \rceil / \tau_0}$, we guarantee

that each estimation satisfies $(\alpha, \tau_0/2\lceil \log_2\left(\frac{\sqrt{2}}{2\theta}\right)\rceil)$-RDP. By composition, the output $r$ satisfies $(\alpha, \tau_0)$-RDP. □

---

**Algorithm 4** GoodRadius (Nissim et al., 2016)

---

1: **Input:** Database $\mathcal{P} \in (\mathcal{X}^d)^n$, desired ball volume $t$, noise multiplier $\sigma_0$, tolerance $\theta$. (We set $\theta = 0.1$ in our experiments.)
2: **Notation:** For a radius $r \geq 0$ and a point $p \in \mathcal{X}^d$, let $\mathcal{B}_r(p)$ denote the number of input points contained in a $\ell_2$-ball of radius $r$ around $p$. That is, $\mathcal{B}_r(p) = \{i : \|x_i - p\|_2 \leq r\}$. Denote $\bar{\mathcal{B}}_r(p) = \min\{\mathcal{B}_r(p), t\}$.
3: For $r \geq 0$, define $L$ as $L(r) = \frac{1}{t} \max_{distinct\ i_1,\ldots,i_t \in [n]} \left\{ \bar{\mathcal{B}}_r(x_{i_1}) + \ldots + \bar{\mathcal{B}}_r(x_{i_t}) \right\}$
4: $r \leftarrow \text{DPBinarySearch}(L, t, \sigma_0, \theta)$
    ▷ *Search for an $r$ s.t. (i) $L(r) \geq t$, (ii) $L(r/2) < t$, (iii) satisfies $(\alpha, \tau_0)$-RDP*
5: **return** $r$

---

**Algorithm 5** DP Binary Search

---

1: **Input:** Function $L$, desired ball volume $t$, noise multiplier $\sigma_0$, tolerance $\theta$.
2: **Initialize:** $r_{\text{low}} \leftarrow 0$ and $r_{\text{high}} \leftarrow \sqrt{2}/2$.
3: $\sigma_0 \leftarrow \sqrt{\alpha \lceil \log_2\left(\frac{\sqrt{2}}{2\theta}\right)\rceil / \tau_0}$
4: **while** $r_{\text{high}} - r_{\text{low}} > \theta$ **do**
5:    $r_{\text{mid}} \leftarrow (r_{\text{low}} + r_{\text{high}})/2$
6:    $\tilde{L}(r_{\text{mid}}) \leftarrow L(r_{\text{mid}}) + \mathcal{N}(0, 4\sigma_0^2)$
7:    $\tilde{L}(r_{\text{mid}}/2) \leftarrow L(r_{\text{mid}}/2) + \mathcal{N}(0, 4\sigma_0^2)$
8:    **if** $\tilde{L}(r_{\text{mid}}/2) \geq t$ **then**
9:       $r_{\text{high}} \leftarrow r_{\text{mid}}$
10:    **else if** $\tilde{L}(r_{\text{mid}}) \geq t$ **then**
11:       $r_{\text{high}} \leftarrow r_{\text{mid}}$
12:    **else**
13:       $r_{\text{low}} \leftarrow r_{\text{mid}}$
14:    **end if**
15: **end while**
16: **return** $(r_{\text{low}} + r_{\text{high}})/2$

---

## B  PRIVACY ANALYSIS

In this section, we provide the privacy analysis for Algorithm 1.

**Theorem 3** (Restatement of Theorem 1). *Algorithm 1 is $(\varepsilon, \delta)$-differentially private.*

We introduce some concepts and relevant theorems from the literature required for our analysis first. While $(\varepsilon, \delta)$-DP is a useful definition of privacy, it does not allow us to tightly quantify the privacy guarantees from the composition of multiple mechanisms. Instead, the notion of Rényi Differential Privacy (RDP) (Mironov, 2017) provides a succinct way to monitor the privacy costs from the composition of multiple mechanisms.

**Definition 2** (Rényi Divergence (Mironov, 2017)). For two probability distributions $P$ and $Q$ defined over $\mathbb{R}$, the Rényi divergence of order $\alpha > 1$ is

$$D_\alpha(P\|Q) = \frac{1}{\alpha - 1} \log \mathbb{E}_{x\sim Q}\left(\frac{P(x)}{Q(x)}\right)^\alpha.$$

**Definition 3** (Rényi Differential Privacy (Mironov, 2017)). A randomized algorithm $\mathcal{A}$ is $\tau$-Rényi differentially private of order $\alpha$ $((\alpha, \tau)$-RDP) if for any two neighboring inputs $\mathcal{X}$ and $\mathcal{X}'$, which differ in only a single record, we have

$$D_\alpha(\mathcal{A}(\mathcal{X})\|\mathcal{A}(\mathcal{X}')) \leq \tau.$$

**Theorem 4** (RDP Sequential Composition (Mironov, 2017)). *If $\mathcal{A}_1$ and $\mathcal{A}_2$ are $(\alpha, \tau_1)$-RDP and $(\alpha, \tau_2)$-RDP respectively then the mechanism combining the two $g(\mathcal{A}_1(\mathcal{X}), \mathcal{A}_2(\mathcal{X}))$ is $(\alpha, \tau_1 + \tau_2)$-RDP.*

**Theorem 5** (RDP to DP conversion (Balle et al., 2020)). *If a mechanism $M$ is $(\alpha, \tau)$-RDP then it is $(\tau + \log((\alpha - 1)/\alpha) - (\log \delta + \log \alpha)/(\alpha - 1), \delta)$-DP for any $0 < \delta < 1$.*

One of the most widely used mechanisms to guarantee RDP is the Gaussian mechanism.

**Theorem 6** (Gaussian Mechanism). *The Gaussian mechanism $M : \mathcal{X} \to \mathbb{R}^m$ of the form*

$$M(x) = q(x) + \mathcal{N}\left(0, \frac{\Delta_2(q)^2 \alpha \mathbf{I}_m}{2\tau}\right)$$

*satisfies $(\alpha, \tau)$-RDP, where $\Delta_2(q) = \max_{x,x'} \|q(x) - q(x')\|_2$ is the $\ell_2$-sensitivity of the query $q$.*

We also need the following privacy amplification theorem by subsampling.

**Theorem 7** (Amplification by subsampling (Wang et al., 2019)). *Given a dataset of $n$ points drawn from a domain $\mathcal{X}$ and a mechanism $\mathcal{M}$ that takes an input from $\mathcal{X}^m$ for $m \leq n$, let the randomized algorithm $\mathcal{M} \circ \text{subsample}$ be defined as: i) subsample without replacement $m$ datapoints of the dataset (sampling parameter $\gamma = m/n$ ), and ii) apply $\mathcal{M}$ to the subsampled dataset. For all integers $\alpha \geq 2$, if $\mathcal{M}$ obeys $(\alpha, \tau(\alpha))$-RDP, then the subsampled mechanism $\mathcal{M} \circ \text{subsample}$ obeys $(\alpha, \tau'(\alpha))$ RDP where,*

$$\tau'(\alpha) \leq \frac{1}{\alpha - 1} \log \left( 1 + \gamma^2 \binom{\alpha}{2} \min \left\{ 4 \left( e^{\tau(2)} - 1 \right), \right. \right.$$
$$e^{\tau(2)} \min \left\{ 2, \left( e^{\tau(\infty)} - 1 \right)^2 \right\} \right\}$$
$$\left. + \sum_{j=3}^{\alpha} \gamma^j \binom{\alpha}{j} e^{(j-1)\tau(j)} \min \left\{ 2, \left( e^{\tau(\infty)} - 1 \right)^j \right\} \right).$$

We are ready to do the privacy analysis of our algorithm (Theorem 1).

*Proof.* Fix one iteration of the algorithm and let us bound the privacy loss. Our algorithm consists of 3 components: a DP algorithm GoodRadius (Line 5), DP projected mean estimation using Gaussian DP Mechanism (Line 20) and DP radius coverage check using Gaussian DP Mechanism (Line 12). The output of GoodRadius algorithm $r$ on Line 5 satisfies $(\alpha, \tau_0)$-RDP by Theorem 2. Consider projected mean estimation using Gaussian Mechanism. The projection operation in Line 19 ensures that the $\ell_2$-sensitivity of the term $\sum_{i=1}^{M} \tilde{p}_{\text{priv}}^i$ in Line 20 is at most $2R$. As we are adding Gaussian noise sampled from $\mathcal{N}(0, 4R^2\sigma_1^2 I)$, we can ensure $\tilde{p}_{\text{priv}}$ in Line 20 satisfies $(\alpha, \tau_1)$-RDP by setting $\sigma_1 = \sqrt{\alpha/2\tau_1}$ (Theorem 6). For the radius coverage check using Gaussian Mechanism, in Line 11, $t$ counts the number of points of $p_{\text{priv}}^1, \ldots, p_{\text{priv}}^M$ covered in a ball with center $\tilde{p}_{\text{priv}}$ and radius $r + \frac{2\lambda R\sigma_1 \sqrt{K}}{M}$. The sensitivity of the term $t$ is 1. Then in Line12 we add Gaussian noise sampled from $\mathcal{N}(0, \sigma_2^2 I)$ to $t$. By setting $\sigma_2 = \sqrt{\alpha/2\tau_2}$, we can show the radius coverage check satisfies $(\alpha, \tau_2)$-RDP (Theorem 6). Our algorithm consists of one instance of $(\alpha, \tau_0)$-RDP algorithm GoodRadius (Line 5), at most $(\hat{T} + 1)$ instances of $(\alpha, \tau_1)$-RDP projected mean estimation using Gaussian DP Mechanism (Line 8 and Line 20) and at most $\hat{T}$ instances of $(\alpha, \tau_2)$-RDP radius coverage check using Gaussian DP Mechanism (Line 12). We apply the composition property of RDP (Theorem 4) to ensure each iteration of the algorithm is $(\alpha, \tau)$-RDP, that is, we ensure that $\tau = \tau_0 + (\hat{T} + 1)\tau_1 + \hat{T}\tau_2$.

Further, note that in each iteration, the Next Token Generation subroutine (Tang et al., 2023) involves drawing $MN$ samples from the dataset $\mathcal{D}_{\text{priv}}$, there is a privacy amplification by subsampling at each iteration. We apply Theorem 7 to show that the effective privacy loss per iteration of our algorithm is $(\alpha, \tau')$-RDP for the full dataset.

Finally, we compose privacy loss across all the $T_{max}$ iterations of our algorithm using composition theorems (Theorem 4). We conclude that Algorithm 1 satisfies $(\alpha, T_{max}\tau')$-RDP and convert the privacy guarantee back into the standard DP definition (Theorem 5). □

We then provide intuition on how the privacy budget is allocated below.

The GoodRadius step initializes a reference radius for the projected ball. Since it is not the final radius, this step can tolerate higher noise levels. Therefore, we typically set $\sigma_0$ to a large value (e.g., 10) to save privacy budget for the more sensitive projected mean estimation step.

The radius coverage check verifies whether a sufficient number of the original probability vectors $p_{\text{priv}}^1, \ldots, p_{\text{priv}}^M$ lie within a small radius from $\tilde{p}_{\text{priv}}$. This involves counting the number of covered vectors, adding Gaussian noise with standard deviation $\sigma_2$, and comparing the noisy count to the total number of vectors $M$. When $M$ is large, the relative impact of the noise added to the count diminishes because the noise becomes negligible compared to $M$. Therefore, as $M$ increases, we can set $\sigma_2$ to a larger value. For example, when $M = 40$, $\sigma_2$ can be set to around 5 without significant loss of accuracy.

In conclusion, our privacy budget allocation prioritizes the projected mean estimation step for high accuracy, while allowing other steps to tolerate higher noise.

## C  EXPERIMENTAL SUPPLEMENTARY

### C.1  EXPERIMENTS COMPUTE RESOURCES

The experiments use 4 NVIDIA RTX A5000 GPUs, each equipped with 24,564 MiB of memory. Approximately 24 hours are required to reproduce the main results in Section 5.1.

### C.2  DATASETS

In this section, we describe the datasets used in our experiments.

- **AGNews.** The AG News (AG's News Corpus) dataset (Zhang et al., 2015) comprises news articles categorized into four labels: World, Sports, Business, and Sci/Tech. It includes 30,000 training samples and 1,900 test samples per class. For our experiments, we randomly select 1,000 samples from the test set.

- **DBPedia.** The DBPedia ontology classification dataset (Zhang et al., 2015) includes contents categorized into one of 14 topics: Company, School, Artist, Athlete, Politician, Transportation, Building, Nature, Village, Animal, Plant, Album, Film, and Book. This dataset includes 40,000 training samples and 5,000 test samples for each class. For our experiments, we randomly select 49,999 samples from the training set and 1,000 samples from the test set.

- **TREC.** The Text REtrieval Conference (TREC) question classification dataset (Voorhees and Tice, 2000) contains questions categorized into one of 6 answer types: Number, Location, Person, Description, Entity, and Abbreviation. The dataset includes 5,452 training samples and 500 test samples, distributed non-uniformly across the categories.

- **MIT Movies.** MIT Movies trivia10k13 dataset (Liu et al., 2012) comprises movie reviews designed for information extraction tasks, with specific slots for movie genre (MIT-G) and director name (MIT-D). For training and testing, MIT-G includes 2,953 and 780 samples respectively, while MIT-D contains 1,561 training samples and 415 test samples.

### C.3  NON-PRIVATE BASELINES

In this section, we present the performance of non-private baselines for $\varepsilon = \infty$ in Table 5: (i) DP few-shot generation algorithm operates without any added noise; (ii) 4-shot demonstrations randomly selected from the private dataset.

### C.4  HYPERPARAMETER SEARCH

In this section, we conduct ablation studies on key hyperparameters in Algorithm 1. We present results for datasets with varying $\lambda$ and $\hat{T}$ values in Table 6-Table 10. Specifically, we consider $\hat{T} \in \{1, 2\}$ and $\lambda \in \{0.15, 0.2, 0.25\}$ and present ICL accuracy for all these hyperparameter choices. We observe that ICL accuracy remains stable across the different hyperparameter settings. For

Table 5: Performance of non-private baselines on test datasets. We report mean and standard deviation of accuracy over 5 runs with different random seeds. $\varepsilon = \infty$ (Algorithm 2 with $\sigma = 0$) column represents DP few-shot generation algorithm operating without any added noise. $\varepsilon = \infty$ (rand) column represents using 4-shot demonstrations randomly selected from the private dataset.

| Task | $\varepsilon = \infty$ (Algorithm 2 with $\sigma = 0$) | $\varepsilon = \infty$ (rand) |
|---|---|---|
| AGNews | $65.92_{1.13}$ | $65.28_{2.65}$ |
| DBPedia | $67.88_{1.75}$ | $69.06_{1.40}$ |
| TREC | $72.32_{2.76}$ | $53.12_{3.72}$ |
| MIT-G | $43.23_{7.00}$ | $42.08_{6.35}$ |
| MIT-D | $78.36_{3.33}$ | $79.33_{1.48}$ |

example, on the AGNews dataset, AdaDPSyn achieves accuracy in the range of 64.28%–65.92% for $\hat{T} \in \{1, 2\}$ and $\lambda \in \{0.15, 0.2, 0.25\}$, compared to 63.18% for DP few-shot generation when $\varepsilon = 8$. This is good in terms of the stability of Algorithm 1.

Table 6: ICL performance on the test dataset of AGNews across varying $\lambda$ and $\hat{T}$ values, showing mean and standard deviation of accuracy over 5 runs with different random seeds. We set $\varepsilon = 8$, $\sigma_0 = 10$, $\sigma_2 = 3$. For specific $\hat{T}$ values, we set $\sigma_1 = 0.58$ when $\hat{T} = 1$, and 0.72 when $\hat{T} = 2$.

| | $\lambda = 0.15$ | $\lambda = 0.2$ | $\lambda = 0.25$ |
|---|---|---|---|
| $\hat{T} = 1$ | $64.28_{1.48}$ | $65.92_{1.02}$ | $64.90_{0.97}$ |
| $\hat{T} = 2$ | $64.38_{3.16}$ | $65.76_{1.13}$ | $65.18_{1.76}$ |

Table 7: ICL performance on the test dataset of DBPedia across varying $\lambda$ and $\hat{T}$ values, showing mean and standard deviation of accuracy over 5 runs with different random seeds. We set $\varepsilon = 8$, $\sigma_0 = 10$, $\sigma_2 = 3$. For specific $\hat{T}$ values, we set $\sigma_1 = 0.73$ when $\hat{T} = 1$, and 0.90 when $\hat{T} = 2$.

| | $\lambda = 0.15$ | $\lambda = 0.2$ | $\lambda = 0.25$ |
|---|---|---|---|
| $\hat{T} = 1$ | $66.14_{2.91}$ | $67.70_{1.68}$ | $66.90_{1.25}$ |
| $\hat{T} = 2$ | $66.02_{3.22}$ | $66.30_{2.09}$ | $65.28_{3.13}$ |

Table 8: ICL performance on the test dataset of TREC across varying $\lambda$ and $\hat{T}$ values, showing mean and standard deviation of accuracy over 5 runs with different random seeds. We set $\varepsilon = 8$, $\sigma_2 = 5$. For specific $\hat{T}$ values, we set $\sigma_0 = 10$ and $\sigma_1 = 0.89$ when $\hat{T} = 1$, and $\sigma_0 = 15$ and $\sigma_1 = 1.09$ when $\hat{T} = 2$.

| | $\lambda = 0.15$ | $\lambda = 0.2$ | $\lambda = 0.25$ |
|---|---|---|---|
| $\hat{T} = 1$ | $70.08_{2.53}$ | $70.28_{2.63}$ | $70.52_{3.62}$ |
| $\hat{T} = 2$ | $71.24_{2.96}$ | $72.08_{2.24}$ | $71.08_{4.31}$ |

Table 9: ICL performance on the test dataset of MIT-G across varying $\lambda$ and $\hat{T}$ values, showing mean and standard deviation of accuracy over 5 runs with different random seeds. We set $\varepsilon = 8$, $\sigma_0 = 10$, $\sigma_2 = 5$. For specific $\hat{T}$ values, we set $\sigma_1 = 0.73$ when $\hat{T} = 1$, and 0.90 when $\hat{T} = 2$.

| | $\lambda = 0.15$ | $\lambda = 0.2$ | $\lambda = 0.25$ |
|---|---|---|---|
| $\hat{T} = 1$ | $37.10_{3.48}$ | $36.33_{7.28}$ | $36.51_{5.03}$ |
| $\hat{T} = 2$ | $36.95_{3.90}$ | $38.31_{5.03}$ | $37.56_{5.47}$ |

We also perform ablation studies on $\sigma_0$ and $\sigma_2$. To evaluate their impact, we conduct experiments on the MIT-G dataset with $\varepsilon = 8$, varying $\sigma_0 \in \{10, 15, 20\}$ and $\sigma_2 \in \{3, 4, 5\}$. The results are presented in Table 11.

Table 10: ICL performance on the test dataset of MIT-D across varying $\lambda$ and $\hat{T}$ values, showing mean and standard deviation of accuracy over 5 runs with different random seeds. We set $\varepsilon = 8$, $\sigma_0 = 15$, $\sigma_2 = 5$. For specific $\hat{T}$ values, we set $\sigma_1 = 0.83$ when $\hat{T} = 1$, and $1.03$ when $\hat{T} = 2$.

|  | $\lambda = 0.15$ | $\lambda = 0.2$ | $\lambda = 0.25$ |
|---|---|---|---|
| $\hat{T} = 1$ | $75.28_{1.93}$ | $74.94_{2.88}$ | $73.30_{2.47}$ |
| $\hat{T} = 2$ | $74.31_{2.32}$ | $74.17_{1.47}$ | $73.11_{2.04}$ |

We observe that ICL accuracy remains stable across different $\sigma_0$ and $\sigma_2$, ranging from $37.08\%$ to $38.41\%$, showing an improvement over the $36.10\%$ achieved by DP few-shot generation. This stability arises because, despite varying $\sigma_0$ and $\sigma_2$, the resulting change in $\sigma_1$ (noise multiplier of projected mean estimation) is very small ($0.90$–$0.92$).

Table 11: ICL performance on the test dataset of MIT-G across varying $\sigma_0$ and $\sigma_2$ values, showing mean and standard deviation of accuracy over 5 runs with different random seeds. We set $\varepsilon = 8$, $\hat{T} = 2$, $\lambda = 0.2$.

|  | $\sigma_0 = 10$ | $\sigma_0 = 15$ | $\sigma_0 = 20$ |
|---|---|---|---|
| $\sigma_2 = 3$ | $37.08_{3.92}$ | $37.87_{4.92}$ | $38.26_{4.55}$ |
| $\sigma_2 = 4$ | $37.15_{3.85}$ | $38.28_{4.30}$ | $38.41_{4.71}$ |
| $\sigma_2 = 5$ | $38.31_{5.03}$ | $37.56_{4.62}$ | $38.18_{4.72}$ |

## C.5 HYPERPARAMETERS

To ensure a fair comparison, we follow the guidance from Tang et al. (2023) to select parameters for the DP few-shot generation algorithm. Specifically, $K$ is held constant at 100, and a grid search is performed to determine the optimal values for $M$ and $N$ ($N \in \{1, 2, 4\}$, $MN \in \{20, 40, 80\}$). We also use the same values of $T_{\max}$ as in Tang et al. (2023) for each task. Hyperparameters for the DP few-shot generation algorithm presented in Table 1 are provided in Table 12. We then use the same values of $M$, $N$, $T_{\max}$ and $K$ for our AdaDPSyn without any additional hyperparameter tuning.

Hyperparameters and privacy parameters for the results presented in Table 2 are provided in Table 13-Table 17.

Table 12: Hyperparameters for DP few-shot generation (Tang et al., 2023) presented in Table 1 and Table 2. We choose $n_{\text{shot}} = 4$ and $K = 100$ for all tasks.

| Task | $M$ | $N$ | $T_{\max}$ | DP |
|---|---|---|---|---|
| AGNews | 10 | 2 | 100 | Gaussian |
| DBPedia | 10 | 2 | 100 | Gaussian |
| TREC | 20 | 2 | 15 | Gaussian |
| MIT-G | 40 | 1 | 20 | Gaussian |
| MIT-D | 40 | 1 | 20 | Gaussian |

Table 13: Hyperparameters and privacy parameters for the main results of AGNews task presented in Table 2. We choose $M = 10$, $N = 2$, $T_{\max} = 100$, $n_{\text{shot}} = 4$ and $K = 100$.

|  | $\hat{T}$ | $\lambda$ | $\sigma_0$ | $\sigma_2$ | $\sigma_1$ |
|---|---|---|---|---|---|
| $\varepsilon = 1$ | 1 | 0.1 | 10 | 3 | 1.23 |
| $\varepsilon = 2$ | 1 | 0.15 | 10 | 3 | 0.92 |
| $\varepsilon = 4$ | 1 | 0.2 | 10 | 3 | 0.71 |
| $\varepsilon = 8$ | 1 | 0.2 | 10 | 3 | 0.58 |

Table 14: Hyperparameters and privacy parameters for the main results of DBPedia task presented in Table 2. We choose $M = 10$, $N = 2$, $T_{\max} = 100$, $n_{\text{shot}} = 4$ and $K = 100$.

|  | $\hat{T}$ | $\lambda$ | $\sigma_0$ | $\sigma_2$ | $\sigma_1$ |
|---|---|---|---|---|---|
| $\varepsilon = 1$ | 1 | 0.1 | 10 | 3 | 1.54 |
| $\varepsilon = 2$ | 1 | 0.1 | 10 | 3 | 1.14 |
| $\varepsilon = 4$ | 1 | 0.2 | 10 | 3 | 0.89 |
| $\varepsilon = 8$ | 1 | 0.2 | 10 | 3 | 0.73 |

Table 15: Hyperparameters and privacy parameters for the main results of TREC task presented in Table 2. We choose $M = 20$, $N = 2$, $T_{\max} = 15$, $n_{\text{shot}} = 4$ and $K = 100$.

|  | $\hat{T}$ | $\lambda$ | $\sigma_0$ | $\sigma_2$ | $\sigma_1$ |
|---|---|---|---|---|---|
| $\varepsilon = 1$ | 1 | 0.1 | 17.5 | 6 | 2.52 |
| $\varepsilon = 2$ | 1 | 0.2 | 15 | 5 | 1.95 |
| $\varepsilon = 4$ | 1 | 0.15 | 10 | 5 | 1.15 |
| $\varepsilon = 8$ | 2 | 0.25 | 15 | 5 | 1.09 |

Table 16: Hyperparameters and privacy parameters for the main results of MIT-G task presented in Table 2. We choose $M = 40$, $N = 1$, $T_{\max} = 20$, $n_{\text{shot}} = 4$ and $K = 100$.

|  | $\hat{T}$ | $\lambda$ | $\sigma_0$ | $\sigma_2$ | $\sigma_1$ |
|---|---|---|---|---|---|
| $\varepsilon = 1$ | 1 | 0.3 | 15 | 6 | 1.59 |
| $\varepsilon = 2$ | 1 | 0.35 | 10 | 6 | 1.17 |
| $\varepsilon = 4$ | 2 | 0.25 | 10 | 6 | 1.12 |
| $\varepsilon = 8$ | 2 | 0.2 | 10 | 5 | 0.90 |

Table 17: Hyperparameters and privacy parameters for the main results of MIT-D task presented in Table 2. We choose $M = 40$, $N = 1$, $T_{\max} = 20$, $n_{\text{shot}} = 4$ and $K = 100$.

|  | $\hat{T}$ | $\lambda$ | $\sigma_0$ | $\sigma_2$ | $\sigma_1$ |
|---|---|---|---|---|---|
| $\varepsilon = 1$ | 1 | 0.15 | 17.5 | 6 | 2.57 |
| $\varepsilon = 2$ | 1 | 0.15 | 17.5 | 6 | 1.49 |
| $\varepsilon = 4$ | 1 | 0.3 | 15 | 6 | 1.07 |
| $\varepsilon = 8$ | 1 | 0.15 | 15 | 5 | 0.83 |

## C.6 EMPIRICAL PRIVACY EVALUATION BY MEMBERSHIP INFERENCE ATTACK

While DP guarantee inherently provides a theoretical guarantee against privacy leakage, it is also important to assess empirical privacy risks. In this section, we evaluate the empirical privacy of AdaDPSyn against membership inference attack (MIA) (Shokri et al., 2017).

We follow the prior work (Duan et al., 2023), where the authors instantiate a membership inference attack (MIA) in the ICL framework. The goal is to determine whether a given private example was used within the prompt. Following Duan et al. (2023), we consider 1-shot ICL. We use the DBPedia dataset and apply the MIA for our AdaDPSyn algorithm. Specifically, we split the dataset into two parts for member and non-member samples. We use member samples to generate 1-shot synthetic demonstration with AdaDPSyn, repeated over 5 trials. Then we attempt MIA for the samples from members and non-members during ICL. For each synthetic demonstration, we run 20 trials for MIA and average across the 100 trials to calculate the AUC. Following Duan et al. (2023), we also consider a non-private baseline $\varepsilon = \infty$ of using actual samples from the private dataset in the prompt. For this baseline, we also average across 100 trials for AUC. The results are presented in Table 18.

We observe that using actual samples from the private dataset leads to successful MIA results (77.37 in the $\varepsilon = \infty$ case). AdaDPSyn reduces MIA AUC to almost random guesses, showing the effectiveness of our method against MIAs.

Table 18: Empirical privacy evaluation for 1-shot ICL by MIA on DBPedia dataset.

| $\varepsilon$ | $\varepsilon = 1$ | $\varepsilon = 2$ | $\varepsilon = 4$ | $\varepsilon = 8$ | $\varepsilon = \infty$ |
|---|---|---|---|---|---|
| MIA AUC | 50.52 | 51.58 | 51.05 | 52.63 | 77.37 |

## C.7 INSTRUCTION-TUNED LLAMA-2-7B RESULTS

In this section, we perform experiments on the instruction fine-tuned Llama-2-7b model. We present the results on DBPedia with $\varepsilon = 4$ using the Llama-2-7b-chat model in Table 19.

We observe that AdaDPSyn outperforms DP few-shot generation and performs close to non-private baselines on the instruction fine-tuned Llama-2-7B-chat model. Additionally, Llama-2-7B-chat achieves better overall performance compared to Llama-2-7B, showing the benefits of instruction fine-tuning.

Table 19: ICL performance on DBPedia with $\varepsilon = 4$ using Llama-2-7b-chat and Llama-2-7b.

| Model | $\varepsilon = 4$ (DP few-shot generation) | $\varepsilon = 4$ (AdaDPSyn) | $\varepsilon = \infty$ (Alg 2, $\sigma = 0$) | $\varepsilon = \infty$ (rand) |
|---|---|---|---|---|
| LLama-2-7B | 64.92 | 67.12 | 67.88 | 69.06 |
| LLama-2-7B-chat | 74.22 | 75.14 | 75.70 | 76.90 |

## C.8 VARYING LLMS OF SIMILAR SCALE

We perform experiments to compare Llama-2-7b and Gemma-7b on DBPedia with $\varepsilon = 4$. The results are presented in Table 20.

We observe that AdaDPSyn outperforms DP few-shot generation and performs close to non-private baselines on both models. Additionally, Gemma-7B achieves better overall performance compared to Llama-2-7B.

Table 20: ICL performance on DBPedia with $\varepsilon = 4$ using Llama-2-7b and Gemma-7b.

| Model | $\varepsilon = 4$ (DP few-shot generation) | $\varepsilon = 4$ (AdaDPSyn) | $\varepsilon = \infty$ (Alg 2, $\sigma = 0$) | $\varepsilon = \infty$ (rand) |
|---|---|---|---|---|
| LLama-2-7B | 64.92 | 67.12 | 67.88 | 69.06 |
| Gemma-7B | 73.54 | 77.14 | 77.36 | 80.64 |

## C.9 VARYING MODEL SIZE

We conduct experiments comparing Llama-2-7b and Llama-2-13b on DBPedia with $\varepsilon = 4$. The results are presented in Table 21. We observe that Llama-2-13B shows better overall performance compared to Llama-2-7B, reflecting the advantages of increased model size.

Table 21: ICL performance on DBPedia with $\varepsilon = 4$ using Llama-2-7b and Llama-2-13b.

| Model | $\varepsilon = 4$ (DP few-shot generation) | $\varepsilon = 4$ (AdaDPSyn) | $\varepsilon = \infty$ (Alg 2, $\sigma = 0$) | $\varepsilon = \infty$ (rand) |
|---|---|---|---|---|
| LLama-2-7B | 64.92 | 67.12 | 67.88 | 69.06 |
| Llama-2-13B | 68.24 | 69.78 | 70.46 | 70.60 |

## C.10 TIME COST ANALYSIS

In this section, we present the time cost for generating one DP synthetic ICL sample with $\varepsilon = 4$ in Table 22.

We observe that the time cost of AdaDPSyn is very close to that of DP few-shot generation. Although AdaDPSyn includes additional steps that contribute to improved ICL accuracy, they do not introduce significant computational overhead.

Table 22: Time cost evaluation for generating one DP synthetic ICL sample with $\varepsilon = 4$.

| Method (seconds) | AGNews | DBPedia | TREC | MIT-G | MIT-D |
|---|---|---|---|---|---|
| AdaDPSyn | 173.2608 | 182.9686 | 43.9833 | 116.7168 | 117.1299 |
| DP few-shot generation | 173.2583 | 182.9660 | 43.9735 | 116.6792 | 117.0923 |

## C.11 PROMPT FORMATS

In this section, we present prompt formats during ICL and the prompt construction functions $PB(\cdot)$ for Next Token Generation (Algorithm 3). While we use the same prompt format during ICL following Tang et al. (2023), we present them here in Table 23 and Table 24 for the convenience of the reader. We also present the prompt construction functions $PB(\cdot)$ used in Next Token Generation (Algorithm 3) in Table 25 and Table 26.

Table 23: The prompts used during ICL for text classification tasks, taken from Table 7 of Tang et al. (2023).

| Task | Prompt | Labels |
|---|---|---|
| AGNews | Classify the news articles into the categories of World, Sports, Business, and Technology.

Article: USATODAY.com - Retail sales bounced back a bit in July, and new claims for jobless benefits fell last week, the government said Thursday, indicating the economy is improving from a midsummer slump.
Answer: Business

Article: New hard-drive based devices feature color screens, support for WMP 10.
Answer: | World, Sports, Business, Technology |
| DBPedia | Classify the documents based on whether they are about a Company, School, Artist, Athlete, Politician, Transportation, Building, Nature, Village, Animal, Plant, Album, Film, or Book.

Article: Geoffrey D. Falksen (born July 31 1982) is an American steampunk writer.
Answer: Artist

Article: The Perrin River is a 1.3-mile-long (2.1 km) tidal river in the U.S. state of Virginia. It is a small inlet on the north shore of the York River near that river's mouth at Chesapeake Bay.
Answer: | Company, School, Artist, Athlete, Politician, Transportation, Building, Nature, Village, Animal, Plant, Album, Film, Book |
| TREC | Classify the questions based on whether their answer type is a Number, Location, Person, Description, Entity, or Abbreviation.

Question: How did serfdom develop in and then leave Russia?
Answer Type: Description

Question: When was Ozzy Osbourne born?
Answer Type: | Number, Location, Person, Description, Entity, Abbreviation |

## C.12 DEMONSTRATIONS

In this section, we present generated samples using AdaDPSyn at $\varepsilon = 4$ for the DBPedia dataset, as shown in Table 27. These results are compared to those obtained using DP few-shot generation, displayed in Table 28. Both methods use the LLama-2-7B-hf model.

We observe that samples generated by AdaDPSyn show good coherence and clarity. For example, the first sample in Table 27, which describes "John P. Haley," offers a well-structured biography that provides relevant details about his political background. Although there are some factual inaccuracies, these errors do not affect the sample's ability to perform well in the classification task. The generated text includes sufficient context and key information to ensure correct classification while maintaining the required privacy guarantees. In contrast, DP few-shot generation shows a clear drop in fluency and coherence. For example, the second sample in Table 28 contains fragmented sen-

Table 24: The prompts used during ICL for information extraction tasks, taken from Table 8 of Tang et al. (2023).

| Task | Prompt |
|---|---|
| MIT-G | Sentence: last to a famous series of animated movies about a big green ogre and his donkey and cat friends
Genre: animated

Sentence: what is a great comedy featuring the talents of steve carell as a loser looking for a friend
Genre: |
| MIT-D | Sentence: in 2005 director christopher nolan rebooted a legendary dc comics superhero with a darker grittier edge in which movie
Director: christopher nolan

Sentence: what 1967 mike nichols film features dustin hoffman in romantic interludes with anne bancroft as mrs robinson
Director: |

Table 25: Prompt construction function $PB(\text{instruction}, \mathcal{D}, y)$ used in Algorithm 3 for text classification tasks, taken from Table 5 of Tang et al. (2023).

| Task | Prompt construction function $PB(\text{instruction}, \mathcal{D}, y)$ | Labels |
|---|---|---|
| AGNews | Given a label of news type, generate the chosen type of news accordingly.

News Type: World
Text: Australia boosts anti-terror measures at small airports SYDNEY: The Australian government announced a major security upgrade for nearly ...

News Type: World
Text: | World, Sports, Business, Technology |
| DBPedia | Given a label of document type, generate the chosen type of document accordingly.

Document Type: Company
Text: Cherry Lane Music was founded in 1960 by Milton Okun in the apartment above the Cherry Lane Theater in Greenwich Village of New York City...

Document Type: Company
Text: | Company, School, Artist, Athlete, Politician, Transportation, Building, Nature, Village, Animal, Plant, Album, Film, Book |
| TREC | Given a label of answer type, generate a question based on the given answer type accordingly.

Answer Type: Number
Text: How many people in the world speak French?

Answer Type: Number
Text: | Number, Location, Person, Description, Entity, Abbreviation |

tences and disjointed phrases, making it difficult to follow. These results show the effectiveness of our adaptive method, which maintains strong privacy guarantees while producing more coherent and usable text compared to DP few-shot generation.

### C.13 COMPARISON OF DP AND REAL SAMPLES

In this section, we compare the DP samples generated by AdaDPSyn with the real private samples. We use the MIT-G dataset for this analysis because this task shows a relatively large performance gap between private and non-private methods. We present the private samples in Table 29, and the DP samples generated using AdaDPSyn with $\varepsilon = 1, 2, 4, 8$ in Table 30.

In the original private samples, the labels (movie genres) are explicitly included in the text, as the task is to extract the genre from the text. Similarly, in the DP samples generated by AdaDPSyn ($\varepsilon = 1, 2, 4, 8$), the labels are included most of the time, highlighted in blue in Table 30. Occasionally, the generated samples use related words instead of the exact label (e.g., "boxer" instead of "boxing"

Table 26: Prompt construction function $PB(\text{instruction}, \mathcal{D}, y)$ used in Algorithm 3 for information extraction tasks, taken from Table 6 of Tang et al. (2023).

| Task | Prompt construction function $PB(\text{instruction}, \mathcal{D}, y)$ |
| --- | --- |
| | Given a genre for the film, generate a description accordingly and make sure to include the given genre in the description. |
| MIT-G | Genre: holiday
Sentence: what is the name of this perennial holiday favorite featuring an elderly miser learning the error of his ways thanks to three ghostly visitations |
| | Genre: action
Sentence: |
| | Given a director for the film, generate a description accordingly and make sure to include the given director in the description. |
| MIT-D | Director: pixar
Sentence: what pixar animated film features a talking dog named dug |
| | Director: disney
Sentence: |

or "basketball" instead of "sports"), highlighted in red in Table 30. Despite these substitutions, the DP samples maintain semantic relevance to the task. However, compared to the private samples, the DP samples are more general and less detailed, sometimes introducing noise or repetition. These differences reflect the trade-offs for ensuring DP. With different $\varepsilon$ values ($\varepsilon = 1, 2, 4, 8$), we do not observe significant differences in the quality of the DP samples. This aligns with our experimental results, where the ICL accuracy shows only small differences across $\varepsilon = 1, 2, 4, 8$ (between $37.41\%$ and $38.31\%$).

We also conduct experiments with smaller $\varepsilon$ (0.2, 0.3, 0.5) on the MIT-G dataset. The DP samples generated using AdaDPSyn with these $\varepsilon$ values are included in Table 30, and we present the performance results in Table 31.

For $\varepsilon = 1, 2, 4, 8$, the ICL performance shows only small differences (between $37.41\%$ and $38.31\%$). However, when $\varepsilon$ decreases to 0.5, ICL accuracy drops to $30.56\%$, and further decreases to $23.75\%$ for $\varepsilon = 0.3$. At $\varepsilon = 0.2$, the ICL accuracy is $13.77\%$, which is nearly identical to the fully private baseline ($\varepsilon = 0$).

In Table 30, we observe that, for $\varepsilon = 1, 2, 4, 8$, the DP samples maintain similar quality, with labels (movie genres) included in the text most of the time. At $\varepsilon = 0.5$, quality decreases with occasional noise, and labels are included less frequently. At $\varepsilon = 0.3$, labels are only occasionally included, and at $\varepsilon = 0.2$, labels are entirely absent, with the samples containing more noise.

Table 27: Examples of generated samples using AdaDPSyn at $\varepsilon = 4$ for DBPedia.

| Demonstration | Label |
|---|---|
| John P. Haley (born July 23, 1946) is an American politician and attorney who served as a Democratic member of the U.S. House of Representatives from 1997 to 2003 representing the 4th District of Virginia. He was a candidate for Governor of Virginia in 2005. He was a candidate for the Democratic nomination for the U.S. Senate in 2006. | Politician |
| The 1899-1900 YMCA Building is a historic building located at 121 South Main Street in downtown Louisville Kentucky. It was built in 1899-1900 and is a three-story brick building with a mansard roof. It is an example of the Richardsonian Romanesque style. It was designed by Louisville architect Henry Whitestone. The building was added to the National Register of Historic | Building |
| The genus Echium includes about 60 species of flowering plants in the family Boraginaceae. The genus is native to the Mediterranean region, Macaronesia, and Africa. Echium species are annual or perennial herbs or shrubs, some of which are cultivated as ornamental plants. They are characterized by showy flowers with a prominent, tubular corolla and two-lipped labellum. The fruit is a dry caps | Plant |
| Lena Malkus (born 1985) is a Finnish female professional golfer. She plays on the Ladies European Tour. She was the Finnish national champion in 2007. She won the 2009 Finnish Ladies Masters. She won the 2010 Finnish Ladies Masters. She won the 2011 Finnish Ladies Masters. She won the 2012 Finnish Ladies Masters | Athlete |
| John Adams (October 30, 1735 – July 4, 1826) was an American statesman and Founding Father who served as the first Vice President (1789–1797) and second President of the United States (1797–1801). He was a leader of American independence in 1776, and served as the first American minister to the new nation of France from 1 | Politician |
| The house was designed by architect Frank Lloyd Wright and built in 1952 for Edgar J. and Liliane Kaufmann in the community of Mill Run Pennsylvania. The home, which sits on the side of a hill overlooking Bear Run in the Laurel Highlands region of southwestern Pennsylvania, is made of reinforced concrete and features a 150-foot (46 m) long cantilevered section, which was the longest cantilever of | Building |
| Vegetable oils are triglycerides extracted from plants. a broad term for fats and oils, including sesame oil, coconut oil, palm oil, and soybean oil. They are liquid at room temperature, and, unlike animal fats, they do not solidify in the refrigerator. Vegetable oils are liquids at room temperature, and, unlike animal fats, they do not solidify in the refrigerator | Plant |
| John McVeigh (born 3 March 1990) is an Australian cricketer. A right-handed batsman and occasional right-arm medium fast bowler, he plays for New South Wales in the Sheffield Shield. He made his debut in first-class cricket for New South Wales against South Australia at the Sydney Cricket Ground on 3 January 2012. McVeigh was part of the New South | Athlete |
| John Adam Smith (born 1952) is a Canadian politician who was the 17th Premier of Prince Edward Island from 1993 to 1996. He was a member of the Legislative Assembly of Prince Edward Island from 1986 to 1996. He was a member of the Liberal Party of Prince Edward Island. He was a member of the Legislative Assembly of Prince Edward Island from 1986 to | Politician |
| The home of the modern... city the fastest growing city in the United States It is the single largest cultural and entertainment center in the Chicago metropolitan area. While the city has few topographic features like hills or rivers some of its outlying communities do have local landmarks. The city is laid out in a grid pattern with wide Chicago Avenue as the geographic center. The Chicago River runs through the city past the Chicago Merchandise Mart and the Chicago Board of Trade This river is | Building |
| 6 plants with blue flowers bloom from June to October. Flowers are
small.
Some other species are in a flower market on <time>today< /time>.
I can take any bus. I don't think a tour. My sister goes to the tour of the flower market by bike. She's back soon. She bought three plants to my family. I hope her tour is going well. They were beautiful. The blue petals | Plant |
| [[ "David_Whitten" │ David Whitten]] (born 2 months ago) is a record-breaking South African sportsperson. He is the three-time Formula One World Champion, and the first European to win the European Tour. He is also the first motorcycle racer to win the Suzuki Australian Superbike Championship. He is the sportsperson with the most Grand Slam titles, and the first to win the Race of Champions. He is | Athlete |

Table 28: Examples of generated samples using DP few-shot generation at $\varepsilon = 4$ for DBPedia.

| Demonstration | Label |
|---|---|
| 'Nic'l Kozloff, born March 12, 1975 in San Francisco, California, holds political views generally in accordance with his political ideological affiliatin with the Democratic Party.. Kozloff attended high school at San Francisco University High School. After attending University of California, Berkeley he earned a bachelor's degree in history from Stanford with a concentration in political science. The 28yr -old is the current | Politician |
| The foundation [imgref tag2 doc], pausing through these great lengths of limestone led more men to gather still! Upon the structure like leakers floord celephaiden stepped between four. Many words it means i d been taken not and some a place inside when only is was something strange within she kept there or only into many time his would fall would reach those upon at you what as him back will help your at their no how may with just did upon which where out these | Building |
| [Name of a plant] is tall with a trunk diameter of up to 30 inches, a canopy diameter of up to 50 feet. The plant grows up to 20 m (66 ft) and is a perennial, deciduous plant. It grows well in a range of climates from the temperate zones in eastern and western North America. It can tolerate drought. The tree can live 300 years. It grows | Plant |
| Olympic compettessor are : Milt Stahmetze the topmost score were was - DUHSHA the Olympics score on him , i and even all , other men for was + is me score will like one at each he my other as two have no do can what it ACT on have ever, two time been him when first last with has what three year no how then you , this they I i this . were + but she - their other can how time a | Athlete |
| 6 politcian found (6. The text for your candidate(economiser and conserv) could 4) in that. So vote and economical! If this article. But i <0xE2> like them as good in parliament too so do what yi have (you could get better one(and then he. Is so many candidates you might also try other type, which are as politica in <span: The only two candidate with name politci <8, <x6C | Politician |
| Buildings make up the most amount of energy use in buildings and account for more than 40% of energy used for commercial, public and institutional uses, and nearly 12% are used to power the homes in which we live. Buildings also contribute 30 to 40% of greenhouse gas emissions and consume a lot of materials such as concrete, steel, plastic, aluminum, wood, and copper. Buildings use energy and materials, | Building |
| Lumo, Lumo is an adherent plant with thin stems, oval leaves and yellow flowers. As the leaves turn red they are eaten by squirrels, and this stimulates the plant to produce flowers. Lumo is an endemic of South and North-Eastern Kenya. Lumo is used to treat stomach ulcers in Kenya. The flowers are eaten to reduce blood pressure. It is also used to so | Plant |
| 7. The Olympics take up one entire page for one individual (and maybe several people together.) For that event we get more people from our audience'. They don&r'ts; come. This would lead more visitors coming over time which makes money (proft for a magazine.). But what will also work with your document and you could see what I wrote on the next section to learn to see and feel like they could help people, to find your information for that one particular subject of | Athlete |
| Internal (Ind vsOuti . . . .. and evopt): For public speaker whose presentation need it by [<fruit flavid = indemonde..r1 <foontime aet4 tusal ] then try .. dts are useful things especially things make fntions gop on at its location bocre than (co/ceofonon rtrttirit...'teefft') by public oers can know <info ∫2 | Politician |
| House [the ] Linden Farm Maitree in Vicks Fakauala Kibila Township; where land records keep register or give documentation cert with witness according at Police'2[...snidifoo bend](-Kotwela): Mr Prilminary officer by OD at ISA G1& TPS Mr Sauman Saulu APC : Deputial Suphanahanepoa from his P1 or original I can witness(this man with p | Building |
| One has given ten the key whereinto these caster [chaps may eat any three food any nine parts must hold]. Also be noted down into nin or for eleven will return not twenty may turn fifteen when 8 not go more any any into turn go twelve this up some back many so thrite if I said more as into but or< a in which have at you must eat 0 I shall it with seven from down with at be < i eat [have into | Plant |
| C-sharp expert Tuan Lee brings professional perspective to Azure architecture. Mr. Lee had originally only viewed using PaaS as a place to rent/ buy Windows , server administration ... read more or manage Linux Web developers prefer web to have, server expert may instead for having something (if don't server available; do or as their for data are managing some files more use databases will them they think not all documents server more... For can access control manage only then should people these | Athlete |

Table 29: Real private samples from MIT-G.

| Real Private Sample: Text (Label) |
| --- |
| what 2011 dennis dugan romantic comedy starred adam sandler jennifer aniston nicole kidman and brooklyn decker (romantic comedy) |
| name the 1993 epic movie starring liam neeson as a german businessman that saved over a thousand polish jewish refugees during the holocost (epic) |
| what is the classic boxing film that starred and was written by superstar sylvester stallone (boxing) |
| what 2001 fantasy film based on a novel by j r r tolkien tells the story frodo baggins (fantasy film) |
| name the disney live action film based upon a classic mickey cartoon where mickey becomes a magician (live action) |
| in 2011 what raunchy comedy was released with four friends attempting to have a bachelor party in thailand where nothing seems to go right i m thinking of a movie where star character stu ed helms wakes up with a tattoo on his face made popular by mike tyson (raunchy comedy) |
| what 1998 steven spielberg war film starred tom hanks matt damon edward burns and tom sizemore (war) |
| what is the animated family movie about a man who has to look after three orphaned girls (animated family) |
| name the sports movie about a high school basketball team that starred gene hackman as the coach (sports) |
| in which animated movie did steve carell play a thief who is trying to steal the moon (animated) |
| what 1956 science fiction film concerns aliens in a small california town who inhabit citizens bodies (science fiction) |
| what 2007 biopic is about the story of jean dominique bauby and his trouble with locked in syndrome (biopic) |
| in a decrepit south american village men are hired to transport an urgent nitroglycerine shipment without the equipment that would make it safe is the plot of this 1953 action thriller (action thriller) |
| young traveler allan gray discovers evidence of the supernatural in this black and white 1932 horror film (horror) |
| this computer animated film stars johnny depp as a timid chameleon who finds courage in the desert (computer animated) |
| what was the name of the cg movie with johnny depp as a lizard in the desert (cg) |
| name the 1990 american romantic fantasy film starring patrick swayze demi moore tony goldwyn and whoopi goldberg (american romantic fantasy) |
| what 2002 brazilian movie had a television show spin off called city of men in 2003 (brazilian) |
| do you know the name of the comedy film starring an actor named michael that s based off of a video game (comedy) |
| what is the 1989 comedy drama crime film starring tom hanks and beasley the dog (comedy drama crime) |
| a classic childrens tale of a bear who enjoys his honey and his animal friends (childrens) |
| superhero movie where team of superheros get together to defeat threats too big for just one hero (superhero movie) |
| what is the superhero movie about a powerful but arrogant warrior who is cast out of asgard (superhero) |
| tim burton used stop motion animation to tell the story of jack skellington in what 1993 disney movie (stop motion animation) |
| what s the animated film made up of several stories which are set to classical music (animated) |
| this movie musical features two rival gangs the sharks and the jets and lots of prancing and snapping and wailing for maria (musical) |
| what is the 1992 american horror film starring tony todd as the titular character who has a hook for a hand (american horror) |
| this one of the first classic horror movies thats been remade multiple times about zombies (classic horror) |

Table 30: Generated DP samples using AdaDPSyn on MIT-G.

| $\varepsilon$ | DP Sample: Text (Label) |
|---|---|
| $\varepsilon = 0.2$ | For Mary Baratani—smart, snay beautiful Miamity born who lost six sizes (romantic comedy) 
 Sentence after my changes."Here some time...more >,"</bold)>in London "before (epic) 
 Vancouver Wow<0xE5>feild! For long now Tito Tararanilla may give (boxing) 
 <0xED>He picked up rocks (but failed [intonation key! R8)) &1 (live action) 
 When Harry becomes Dr. Audi McAr's room guest but isn't expected again (raunchy comedy) 
 With bombs clancing wildly an man a a few fell foming . man cming together (war) 
 (for that in its purpose from something makes necessary have such ability will like must (an inter, (alien film) 
 <0xEB> Anything where she wins ... has better movie written . One fine episode & everyone clapend (animated family) 
 (Example above replaced bengaleafcricket match nether on hrs: match win/ (sports) 
 ... he started exploring areas forbidden throughout each human group except there ended up being definitely less area (animated) |
| $\varepsilon = 0.3$ | For Mary, the path to true love romantic comedy took a weird turn when she unw (romantic comedy) 
 199BCE, the Persian Empire attack the Greek city-states. Epic cinemat (epic) 
 1976 sports drama film boxing, directed by Tariq Nashe (boxing) 
 199... movies called Rocky because rocky is a boxer and uh they (live action) 
 90s raunchy comedy remake of the 1980s classic comedic (raunchy comedy) 
 1917 is a war film film filled with action and also intrigue. WW1 (war) 
 1996 teen disaster film is about a group of teenagers who are dealing (alien film) 
 <0xEB>is a 201 in the direction of Lanipekun, according to the (animated family) 
 (Example) in the 1990s, the man who trained the wrestler " (sports) 
 ... he started to find its way into the pivotal role of the bromantic comedy. (animated) |
| $\varepsilon = 0.5$ | 2006 romantic comedy about a single boy meeting a girl while they are both visiting (romantic comedy) 
 2006 epic movie about a legendary samurai's fight against Lord K (epic) 
 2010 is a biographical drama film about the life of boxer james j. (boxing) 
 2000 and 2001 a space odys (fantasy film) 
 201 Action and Adventure Films Sentence 4 (live action) 
 2011s Crazy, Stupid, Love is a raunchy comedy which (raunchy comedy) 
 1998 war film Trenchcoat 22037 is not your average (war) 
 201 mins. 201x201x201x2 (animated family) 
 2014 sports movie with shia labeouf and an old-school baseball style (sports) 
 2014 animated film about a cute girl trying to escape a mysterious castle while the (animated) |
| $\varepsilon = 1$ | 2004 romantic comedy about a woman who loses her job and goes to a resort (romantic comedy) 
 1959 epic film about the battle of waterloo starring charlton h (epic) 
 2006 boxing film starring 2006 boxing film starring (boxing) 
 2017 fantasy film about a young boy who discovers a magical world in his (fantasy film) 
 1990s live action film about a young boy who wants to be a firefigh (live action) 
 2011 raunchy comedy about a man who wakes up with a hangover and (raunchy comedy) 
 1998 war film directed by and starring Clint Eastwood. The film is based (war) 
 1998 animated family film about a young boy who finds a magical board game that trans (animated family) 
 1994 sports movie about a football player who loses his eye in a car accident and (sports) 
 1998 animated film about a boy who dreams of becoming a rock star and the road (animated) |
| $\varepsilon = 2$ | 2004 romantic comedy starring jennifer aniston and adam sandler (romantic comedy) 
 1959 epic film about the battle of waterloo starring charlton h (epic) 
 2005 film about a boxer who is the son of a boxer who was killed (boxing) 
 2017 fantasy film about a young boy who is transported to a magical world (fantasy film) 
 1990s live action film about a young boy who is taken to a magical land (live action) 
 2011 raunchy comedy about a man who wakes up in a hotel room with (raunchy comedy) 
 1998 war film directed by and starring Clint Eastwood. The film is based (war) 
 2016 animated family film about a young boy who wants to be a hero and his journey (animated family) 
 2014 film about a high school basketball team in the 1950s that (sports) 
 2010 animated film about a boy who gets a magic crayon that can draw anything (animated) |
| $\varepsilon = 4$ | 2005 romantic comedy starring jennifer aniston and vince vaug (romantic comedy) 
 1999 epic film about a young boy who is the son of a king and is (epic) 
 1976 film about a boxer who is trying to make it to the top of the (boxing) 
 2017 fantasy film directed by 2017 fantasy film directed by (fantasy film) 
 1999 live action film about a young boy who is transported to a magical world (live action) 
 2011 raunchy comedy about a man who is trying to get his life back on (raunchy comedy) 
 1942 war film directed by howard hawks and starring humphrey bog (war) 
 2010 animated family film about a young boy who befriends a dragon and helps (animated family) 
 2010 was a great year for sports movies, with the release of the 2 (sports) 
 2016 animated film about a boy who is a wizard and his pet dragon. (animated) |
| $\varepsilon = 8$ | 2009 romantic comedy starring jennifer aniston and jason bateman (romantic comedy) 
 1999 epic film about a young man who goes on a journey to find his father (epic) 
 1976 film about a boxer who is trying to make it to the top of the (boxing) 
 2017 fantasy film about a young boy who is transported to a magical world (fantasy film) 
 1999 live action film about a young boy who is transported to a magical world (live action) 
 2011 raunchy comedy about a man who is trying to get his life back on (raunchy comedy) 
 1917 is a war film about two soldiers who are sent on a mission to deliver a (war) 
 2010 animated family film about a young boy who is transported to a magical world (animated family) 
 2010 was a great year for sports movies, with the release of the 2 (sports) 
 2016 animated film about a boy who is transported to a magical world of g (animated) |

Table 31: ICL performance on MIT-G using AdaDPSyn.

| $\varepsilon = 0$ | $\varepsilon = 0.2$ | $\varepsilon = 0.3$ | $\varepsilon = 0.5$ | $\varepsilon = 1$ | $\varepsilon = 2$ | $\varepsilon = 4$ | $\varepsilon = 8$ |
|---|---|---|---|---|---|---|---|
| 13.62 | 13.77 | 23.75 | 30.56 | 37.59 | 37.41 | 37.85 | 38.31 |

