# OpenReview forum: "Data-adaptive Differentially Private Prompt Synthesis for In-Context Learning"
_ICLR.cc/2025/Conference — ICLR 2025 Poster_

### Official Review · Reviewer_BThb · 2024-10-29

**Soundness:** 3
**Presentation:** 3
**Contribution:** 2
**Rating:** 6
**Confidence:** 4

**Summary:**

This work introduces AdaDPSyn to safeguard sensitive data used in LLM prompts in the ICL process. A hypothesis that the next-token predictions by different examples approximately reach a consensus is verified and used to motivate the DP guaranteed in-context sample synthesis in an adaptive manner.

**Strengths:**

This paper developed a dynamic method to obfuscate real private samples to DP guaranteed private sample from being leaked by the LLM during ICL. They dynamic radius update for privately aggregation is interesting although the idea of replace real sample with its DP version is widely applied.

**Weaknesses:**

1. An amount of M queries needs to be made for synthesizing one DP in-context sample. This is of high computation cost. I would like to see some of the computational cost analysis, including the time cost comparison with baselines used in the paper.
2. The comparison of generated DP samples and real private samples are not adequate. Only DP samples of epsilon=4 is shown without direct comparison with private samples. Results of epsilon of other values as well their comparison with private samples should be included.

**Questions:**

I am wondering, sine the goal of this paper is "to mitigate the risk of LLMs potentially leaking private information contained in examples in the prompt", if an example is used as in-context learning samples and sent to the LLMs, why should its privacy issue be considered? The sender party already knows the sample, and the LLM sees the sample. So who should be stopped from seeing the sample?

---

> ### Author Response · Authors · 2024-11-22
> **Responses to Reviewer BThb**
>
> We thank the reviewer for the careful reading and thoughtful comments. We address the reviewer's questions in the following and have revised the paper accordingly. The changes are marked in blue in our revision. We hope the responses below address the reviewer's concerns.
>
> **Q1:** An amount of $M$ queries needs to be made for synthesizing one DP in-context sample. This is of high computation cost. I would like to see some of the computational cost analysis, including the time cost comparison with baselines used in the paper.
>
> **A1:** Thank you for the insightful comment. Both our AdaDPSyn and DP few-shot generation (Tang et al., 2023) require $M$ queries during the synthesis process. The value of $M$ in our experiments is typically small (e.g., 10), which keeps the computational cost manageable. Additionally, the rapid advancements in LLM inference acceleration [1,2,3] suggest that querying LLMs may involve much lower time costs in the near future.
>
> We appreciate the reviewer’s suggestion for a time cost comparison. We present the time cost for generating one DP synthetic ICL sample with $\varepsilon=4$ in the table below.
>
> | Method (seconds)  | AGNews | DBPedia | TREC | MIT-G | MIT-D |
> |---|---|---|---|---|---|
> | AdaDPSyn | 173.2608 | 182.9686 | 43.9833  | 116.7168 | 117.1299 |
> | DP few-shot generation | 173.2583 | 182.9660 | 43.9735 | 116.6792 | 117.0923 |
>
> We observe that the time cost of AdaDPSyn is very close to that of DP few-shot generation (Tang et al., 2023). Although AdaDPSyn includes additional steps that contribute to improved ICL accuracy, they do not introduce significant computational overhead.
>
> We have included this discussion in the updated draft (Appendix C.10). Thank you for highlighting this important point.
>
> [1] Cai, Tianle, et al. "Medusa: Simple LLM Inference Acceleration Framework with Multiple Decoding Heads." Forty-first International Conference on Machine Learning.
>
> [2] Zhao, Yao, et al. "Lookahead: An inference acceleration framework for large language model with lossless generation accuracy." Proceedings of the 30th ACM SIGKDD Conference on Knowledge Discovery and Data Mining. 2024.
>
> [3] Li, Jinhao, et al. "Large Language Model Inference Acceleration: A Comprehensive Hardware Perspective." arXiv preprint arXiv:2410.04466 (2024).
>
> **Q2:** The comparison of generated DP samples and real private samples are not adequate. Only DP samples of $\varepsilon=4$ is shown without direct comparison with private samples. Results of epsilon of other values as well their comparison with private samples should be included.
>
> **A2:** Thank you for highlighting the importance of a comparison between the generated DP samples and real private samples. We have addressed this in the updated draft (Appendix C.13). Specifically, we use the MIT-G dataset for this analysis because this task shows a relatively large performance gap between private and non-private methods. We present the private samples in Table 29, and the DP samples generated using AdaDPSyn with $\varepsilon=1,2,4,8$ in Table 30. We state our main observations in the following.
>
> In the original private samples, the labels (movie genres) are explicitly included in the text, as the task is to extract the genre from the text. Similarly, in the DP samples generated by AdaDPSyn, the labels are included most of the time. Occasionally, the generated samples use related words instead of the exact label (e.g., "boxer" instead of "boxing" or "basketball" instead of "sports"). Despite these substitutions, the DP samples maintain semantic relevance to the task. However, compared to the private samples, the DP samples are more general and less detailed, sometimes introducing noise or repetition. These differences reflect the trade-offs for ensuring DP. With different $\varepsilon$ values ($\varepsilon =1, 2, 4, 8$), we do not observe significant differences in the quality of the DP samples. This aligns with our experimental results, where the ICL accuracy shows only small differences across $\varepsilon=1,2,4,8$ (between $37.41\\%$ and $38.31\\%$).
>
> **Q3:** I am wondering, since the goal of this paper is "to mitigate the risk of LLMs potentially leaking private information contained in examples in the prompt", if an example is used as in-context learning samples and sent to the LLMs, why should its privacy issue be considered? The sender party already knows the sample, and the LLM sees the sample. So who should be stopped from seeing the sample?
>
> **A3:** Thank you for the question. We clarify that our framework involves two LLMs: a trusted LLM and an untrusted LLM. In stage 1, we use AdaDPSyn with a trusted LLM, such as an open-source model, to generate DP synthetic ICL examples. In stage 2, we use these examples to prompt an untrusted LLM for ICL. We should stop the untrusted LLM from seeing the private data samples.

---

> > ### Author Response · Authors · 2024-11-22
> > **Thank you!**
> >
> > We thank the reviewer again for the helpful comments and suggestions for our work. If our response resolves your concerns to a satisfactory level, we kindly request the reviewer to consider raising the rating of our work. Certainly, we are more than happy to address any further questions that you may have.

---

> > ### Comment · Reviewer_BThb · 2024-11-23
> >
> > Thanks a lot for your reply. I have one further question for **Q2** and **A2**.
> >
> > It seems that, $\epsilon$ does not have a strong effect on the result in the range of $1,2,4,8$. So I am wondering, would smaller $\epsilon$ (e.g. $\epsilon=0.001, 0.01, ...$) would bring some differences to the performance and the generated DP samples? I would prefer some further experiments on this part.

---

> > > ### Author Response · Authors · 2024-11-24
> > >
> > > Thank you for the insightful comment. We have conducted additional experiments with smaller $\varepsilon$ (0.2, 0.3, 0.5) on the MIT-G dataset. The DP samples generated using AdaDPSyn with these $\varepsilon$ values are included in our updated draft (Table 30), and we present the performance results in table below.
> > >
> > > | $\varepsilon=0$ | $\varepsilon=0.2$ | $\varepsilon=0.3$ | $\varepsilon=0.5$ | $\varepsilon=1$ | $\varepsilon=2$ | $\varepsilon=4$ | $\varepsilon=8$ |
> > > |---|---|---|---|---|---|---|---|
> > > | 13.62 | 13.77 | 23.75 | 30.56 | 37.59 | 37.41 | 37.85 | 38.31 |
> > >
> > > For $\varepsilon=1,2,4,8$, the ICL performance shows only small differences (between $37.41\\%$ and $38.31\\%$). However, when $\varepsilon$ decreases to 0.5, ICL accuracy drops to $30.56\\%$, and further decreases to $23.75\\%$ for $\varepsilon=0.3$. At $\varepsilon=0.2$, the ICL accuracy is $13.77\\%$, which is nearly identical to the fully private baseline ($\varepsilon=0$).
> > >
> > > In Table 30, we observe that, for $\varepsilon =1, 2, 4, 8$, the DP samples maintain similar quality, with labels (movie genres) included in the text most of the time. At $\varepsilon =0.5$, quality decreases with occasional noise, and labels are included less frequently.
> > > At $\varepsilon = 0.3$, labels are only occasionally included, and at $\varepsilon = 0.2$, labels are entirely absent, with the samples containing more noise.
> > >
> > > Thank you again for your feedback, and we are happy to address any further questions you might have.

---

> > > > ### Comment · Reviewer_BThb · 2024-11-25
> > > >
> > > > Thanks a lot for your reply. I have no further question. Besides, if score "7" exits, I would like to increase my score. But sadly, it does not, so I just keep my score.

---

> > > > > ### Author Response · Authors · 2024-11-25
> > > > > **Thank you!**
> > > > >
> > > > > Thank you very much for your positive feedback! Your valuable suggestions have helped us improve the quality of our work. We really appreciate it.

---

### Official Review · Reviewer_cNDS · 2024-11-01

**Soundness:** 3
**Presentation:** 3
**Contribution:** 3
**Rating:** 6
**Confidence:** 4

**Summary:**

This work focuses on mitigating privacy leakage during in-context learning of LLMs. To do this, they take an existing DP ICL framework from Tang et al., 2023 and include a data-adaptive step during private aggregation. The key idea is that the majority of the output distributions can be clustered together via a ball with some radius R which can be utilized as the local sensitivity. The authors scale the Guassian noise with a differentially private estimate of R. Their experimental results demonstrate that this data-adaptive step boosts the utility over the original framework.

**Strengths:**

1. The paper tackles an important problem of preserving privacy during In-Context Learning of LLMs.
2. I believe the direction of data-adaptive techniques to improve privacy-utility tradeoffs of DP algorithms is important for broader adoption of DP
3. Their proposed technique is technically interesting
4. The paper is well-written, and the analysis seems sound. In particular, I appreciate that the authors use the Amplification by subsampling theorem (Wang et al., 2019) in their analysis, which is the correct amplification to use as opposed to Poisson subsampling used by Tang et al., 2023.

**Weaknesses:**

1. Although the data-adaptive technique is novel, the work relies heavily on the framework introduced by Tang et al., 2023. Seemingly, author’s overall contribution is only a modification on top of an existing DP generation framework by adding a data-adaptive technique to the private aggregation step.
2. Their method introduces 4 more parameters, $\hat{T}, \lambda, \sigma_0, \sigma_2$, than Tang et al., 2023, which requires additional hyperparameter tunning on other datasets. Furthermore, the authors do not provide ablation studies on $\sigma_0, \sigma_2$ which makes it difficult to interpret their effect on the privacy-utility tradeoff.
3. I’m a bit confused about some of the experimental setup. Why did the authors use only a pre-trained, open-weight model in their evaluations, but additionally included two instruction fine-tuned, closed-source models in their ablation? Could you provide results for the instruction fine-tuned Llama-2-7b model?
4. What about the zero-shot performance for GPT-3.5 Turbo and GPT-4o mini in the varying LLMs ablation?
5. The ablation studies are insufficient. For varying LLMs, it would be nice to compare LLMs of similar scale, e.g. Llama-2-7b vs gemma-7b. I’m unsure if GPT-3.5 Turbo and GPT-4o mini are even on the same parameter scale as Llama-2-7b. Moreover, another ablation like varying model size similar to Tang et al., 2023 is missing.

**Questions:**

1. What were the center values and how were the center values chosen for the experiments in Table 1?

---

> ### Author Response · Authors · 2024-11-22
> **Responses to Reviewer cNDS (Part 1)**
>
> We thank the reviewer for the careful reading and thoughtful comments. We address the reviewer's questions in the following and have revised the paper accordingly. The changes are marked in blue in our revision. We hope the responses below address the reviewer's concerns.
>
> **Q1:** Although the data-adaptive technique is novel, the work relies heavily on the framework introduced by Tang et al., 2023. Seemingly, author’s overall contribution is only a modification on top of an existing DP generation framework by adding a data-adaptive technique to the private aggregation step.
>
> **A1:** Thank you for recognizing the novelty of our data-adaptive technique. While our work builds on the framework in Tang et al. (2023), it makes a substantial contribution by addressing a critical problem in generating DP synthetic ICL examples: **Optimizing the privacy-utility trade-off by leveraging the statistical properties of the data.** Specifically, we observe that next-token generation probability vectors often cluster tightly within a small ball, motivating a more effective prompt synthesis algorithm. We would like to clarify that **our method is not a minor modification but involves multiple carefully designed steps**, including establishing the target radius using GoodRadius, private projected mean estimation, private radius coverage check, and iterative radius updates. These components collectively enable a novel data-adaptive prompt synthesis method that significantly improves ICL performance. Our experimental results show clear improvements over Tang et al. (2023) and closely approach the performance of non-private ICL baselines. Besides, we believe that such **data-adaptive technique may be applicable in various problems beyond prompt synthesis, thus is of independent interest**. We clarify this point in the revised version as well.
>
> **Q2:** Their method introduces 4 more parameters, $\hat{T}, \lambda, \sigma_0, \sigma_2$, than Tang et al., 2023, which requires additional hyperparameter tuning on other datasets. Furthermore, the authors do not provide ablation studies on $\sigma_0, \sigma_2$ which makes it difficult to interpret their effect on the privacy-utility tradeoff.
>
> **A2:** Thank you for the insightful comment. We address the reviewer’s concern about parameters as follows:
>
> 1. Hyperparameters $\lambda$ and $\hat{T}$: We conduct a hyperparameter search for $\lambda$ and $\hat{T}$ across all studied datasets in Appendix C.4. Specifically, we consider $\hat{T}\in\\{1, 2\\}$ and $\lambda\in\\{0.15, 0.2, 0.25\\}$, and present ICL accuracy for all these hyperparameter values on each dataset when \\(\varepsilon = 8\\). Our results show that ICL accuracy remains stable across different hyperparameter settings. For example, on AGNews, AdaDPSyn achieves accuracy between $64.28\\%$ and $65.92\\%$, compared to $63.18\\%$ for DP few-shot generation. This stability shows that AdaDPSyn can generalize well to other datasets with minimal hyperparameter tuning.
>
> 2. Privacy parameters $\sigma_0$ and $\sigma_2$: These parameters control the allocation of the overall privacy budget, and their selection generalizes well across datasets based on consistent principles:
>
> (1) $\sigma_0$ (noise multiplier of GoodRadius in Line 5): GoodRadius initializes a reference radius for the projected ball. Since it does not determine the final radius, it can tolerate higher noise. We typically set $\sigma_0$ to a large value (e.g., 10) to conserve privacy budget for the projected mean estimation step.
>
> (2) $\sigma_2$ (noise multiplier of radius coverage check in Line 12): The radius coverage check verifies whether a sufficient number of the original probability vectors $p_\text{priv}^1,\ldots,p_\text{priv}^M$ lie within a small radius from $\tilde{p}_\text{priv}$. Specifically, this step counts the number of covered vectors, adds Gaussian noise with standard deviation $\sigma_2$, and compares the noisy count to the total number of vectors $M$. The choice of $\sigma_2$ depends on the value of $M$. When $M$ is large, the relative impact of the noise added to the count diminishes, allowing for larger $\sigma_2$.
>
> We appreciate the reviewer’s suggestion to perform ablation studies on $\sigma_0$ and $\sigma_2$. To evaluate their impact, we conduct additional experiments on the MIT-G dataset with $\varepsilon=8$, varying $\sigma_0 \in \\{10, 15, 20\\}$ and $\sigma_2 \in \\{3, 4, 5\\}$. The results are presented in the table below.
> |   | $\sigma_0 = 10$ | $\sigma_0 = 15$ | $\sigma_0 = 20$ |
> |---|---|---|---|
> | $\sigma_2 = 3$ | 37.08 | 37.87 | 38.26 |
> | $\sigma_2 = 4$ | 37.15 | 38.28 | 38.41 |
> | $\sigma_2 = 5$ | 38.31 | 37.56 | 38.18 |

---

> > ### Author Response · Authors · 2024-11-22
> > **Responses to Reviewer cNDS (Part 2)**
> >
> > We observe that ICL accuracy remains stable across different $\sigma_0$ and $\sigma_2$, ranging from $37.08\\%$ to $38.41\\%$, showing an improvement over the $36.10\\%$ achieved by DP few-shot generation (Tang et al., 2023). This stability arises because, despite varying $\sigma_0$ and $\sigma_2$, the resulting change in $\sigma_1$ (noise multiplier of projected mean estimation) is very small (0.90–0.92). We have added the results to our updated draft (Table 11).
> >
> > **Q3:** I’m a bit confused about some of the experimental setup. Why did the authors use only a pre-trained, open-weight model in their evaluations, but additionally included two instruction fine-tuned, closed-source models in their ablation? Could you provide results for the instruction fine-tuned Llama-2-7b model?
> >
> > **A3:** Thank you for your question. We clarify that both AdaDPSyn and DP few-shot generation (Tang et al., 2023) are designed to work with **any LLM**, as long as API-level access to the LLM is available. In our experimental setup, our primary focus is ensuring a fair comparison between AdaDPSyn and the baselines by using the same LLM, regardless of whether the model is instruction fine-tuned or not. In the ablation studies, we additionally experiment on GPT-3.5 Turbo and GPT-4 mini to show that the performance of downstream tasks can be further improved using more powerful LLMs. For instance, ICL accuracy improves to $93.08\\%$ for $\varepsilon=4$ using GPT-4o mini compared with $67.12\\%$ for Llama-2-7b.
> >
> > We appreciate the reviewer's suggestion to experiment with the instruction fine-tuned Llama-2-7b model. We present the results on DBPedia with $\varepsilon=4$ using the instruction fine-tuned Llama-2-7b-chat model in the table below.
> >
> > | Model  | $\varepsilon=4$ (DP few-shot generation) | $\varepsilon=4$ (AdaDPSyn) | $\varepsilon=\infty$ (Alg 2, $\sigma = 0$) | $\varepsilon=\infty$ (rand) |
> > |---|---|---|---|---|
> > | LLama-2-7B |  64.92 | 67.12 | 67.88  | 69.06 |
> > | LLama-2-7B-chat | 74.22 | 75.14 | 75.70 | 76.90 |
> >
> > We observe that AdaDPSyn outperforms DP few-shot generation and performs close to non-private baselines on the instruction fine-tuned Llama-2-7B-chat model. Additionally, Llama-2-7B-chat achieves better overall performance compared to Llama-2-7B, showing the benefits of instruction fine-tuning.
> >
> > We thank the reviewer for the insightful comment and we have added the results to our updated draft (Appendix C.7).
> >
> > **Q4:** What about the zero-shot performance for GPT-3.5 Turbo and GPT-4o mini in the varying LLMs ablation?
> >
> > **A4:** Thank you for the suggestion. We have included the 0-shot performance for GPT-3.5 Turbo and GPT-4o mini in the table below. The results show that ICL examples, including those generated with AdaDPSyn, improve performance compared to 0-shot setting. We have included the results in the updated draft (Table 4).
> >
> > | Model  | $\varepsilon=0$ (0-shot) | $\varepsilon=4$ (DP few-shot generation) | $\varepsilon=4$ (AdaDPSyn) | $\varepsilon=\infty$ |
> > |---|---|---|---|---|
> > | GPT-3.5 Turbo |  69.50 | 89.94 | 91.92  | 93.26 |
> > | GPT-4o mini | 85.10 | 91.40 | 93.08 | 94.32 |
> >
> > **Q5:** The ablation studies are insufficient. For varying LLMs, it would be nice to compare LLMs of similar scale, e.g. Llama-2-7b vs gemma-7b. I’m unsure if GPT-3.5 Turbo and GPT-4o mini are even on the same parameter scale as Llama-2-7b. Moreover, another ablation like varying model size similar to Tang et al., 2023 is missing.
> >
> > **A5:** Thank you for the insightful comment. We have performed additional experiments to compare Llama-2-7b and Gemma-7b on DBPedia with $\varepsilon=4$. The results are presented in the table below.
> >
> > | Model  | $\varepsilon=4$ (DP few-shot generation) | $\varepsilon=4$ (AdaDPSyn) | $\varepsilon=\infty$ (Alg 2, $\sigma = 0$) | $\varepsilon=\infty$ (rand) |
> > |---|---|---|---|---|
> > | LLama-2-7B |  64.92 | 67.12 | 67.88  | 69.06 |
> > | Gemma-7B | 73.54 | 77.14 | 77.36 | 80.64 |
> >
> > We observe that AdaDPSyn outperforms DP few-shot generation and performs close to non-private baselines on both models. Additionally, Gemma-7B achieves better overall performance compared to Llama-2-7B.
> >
> > We also appreciate the suggestion to include an analysis of varying model sizes. To address this, we compare Llama-2-7b and Llama-2-13b on DBPedia with $\varepsilon=4$. The results are as follows.
> >
> > | Model  | $\varepsilon=4$ (DP few-shot generation) | $\varepsilon=4$ (AdaDPSyn) | $\varepsilon=\infty$ (Alg 2, $\sigma = 0$) | $\varepsilon=\infty$ (rand) |
> > |---|---|---|---|---|
> > | LLama-2-7B |  64.92 | 67.12 | 67.88  | 69.06 |
> > | LLama-2-13B | 68.24 | 69.78 | 70.46 | 70.60 |
> >
> > We observe that Llama-2-13B shows better overall performance compared to Llama-2-7B, reflecting the advantages of increased model size.
> >
> > We have included the results in the updated draft (Appendices C.8 and C.9).

---

> > > ### Author Response · Authors · 2024-11-22
> > > **Responses to Reviewer cNDS (Part 3)**
> > >
> > > **Q6:** What were the center values and how were the center values chosen for the experiments in Table 1?
> > >
> > > **A6:** Thank you for the question. The center of the projected ball is not a fixed parameter but is calculated during the iterative process using the private projected mean of $\tilde{p}^1_\text{priv},\ldots,\tilde{p}^M_\text{priv}$. We outline our Precision-Focused Iterative Radius Reduction technique as follows:
> > >
> > > Given next-token probability vectors $p^1_\text{priv},\ldots,p^M_\text{priv}$, we initialize $\tilde{p}^i_\text{priv}\gets p^i_\text{priv}$, and calculate the initial center as $\tilde{p}\_\text{priv} \gets \frac{1}{M} \left(\sum_{i=1}^M \tilde{p}\_\text{priv}^{i} + \mathcal{N}(0, 4R^2\sigma_1^2I)\right)$. The initial radius is set to $R\gets \sqrt{2}/2$, and a target radius $r$ is determined using GoodRadius as a lower bound for refinement. The iterative refinement process begins by checking whether a sufficient number of probability vectors are still enclosed within a smaller ball (radius coverage check). If this condition is satisfied, $R$ is reduced towards $r$, and we project $p^1_\text{priv},\ldots,p^M_\text{priv}$ onto a ball with the current center $\tilde{p}_\text{priv}$ and the updated radius $R$. After this projection, the private mean of the projected vectors becomes the next center.
> > >
> > > ---
> > >
> > > We thank the reviewer again for the helpful comments and suggestions for our work. If our response resolves your concerns to a satisfactory level, we kindly request the reviewer to consider raising the rating of our work. Certainly, we are more than happy to address any further questions that you may have.

---

> > > > ### Author Response · Authors · 2024-11-25
> > > > **A gentle reminder**
> > > >
> > > > Dear Reviewer cNDS,
> > > >
> > > > We've taken your initial feedback into careful consideration in our response. Could you please check whether our responses have properly addressed your concerns? If so, could you please kindly consider increasing your initial score accordingly? Certainly, we are more than happy to answer your further questions.
> > > >
> > > > Thank you for your time and effort in reviewing our work!
> > > >
> > > > Best Regards,
> > > >
> > > > Authors

---

> > > > > ### Comment · Reviewer_cNDS · 2024-11-25
> > > > > **Response by Reviewer**
> > > > >
> > > > > I thank the authors for their detailed response. They've addressed my concerns about the experimental setup. I still feel the author's work relies heavily on Tang et al., 2023, however, I have more appreciation for the data-adaptive technique given the newly obtained experimental results. So I'm raising my score to 6. Thank you for your effort!

---

> > > > > > ### Author Response · Authors · 2024-11-25
> > > > > > **Thank you!**
> > > > > >
> > > > > > Thank you very much for your positive feedback! Your valuable input has helped us improve the quality of our work significantly.

---

### Official Review · Reviewer_hxch · 2024-11-04

**Soundness:** 3
**Presentation:** 3
**Contribution:** 2
**Rating:** 6
**Confidence:** 4

**Summary:**

The paper presents AdaDPSyn, a novel data-adaptive differentially private algorithm designed to generate synthetic examples from a private dataset for in-context learning (ICL) with large language models (LLMs). To prevent potential leakage of private information in the examples used for ICL, AdaDPSyn adjusts the noise level (introduced by differential privacy) during data synthesis based on the statistical properties of the dataset. This adjustment ensures high ICL accuracy while adhering to differential privacy guarantees. A key innovation is the Precision-Focused Iterative Radius Reduction technique, which dynamically refines the noise aggregation radius by analyzing data clustering patterns, effectively reducing additive noise.

**Strengths:**

S1: Experiments demonstrate that AdaDPSyn outperforms the baseline of 'DP few-shot generation'.

S2: The paper is well-written, and the ideas are easy to follow.

**Weaknesses:**

W1: Only one baseline was considered, specifically "DP few-shot generation."

W2: If I am not mistaken, the iterative aspect is not utilized significantly, as $\hat{T}$ is frequently set to 1.

**Questions:**

Please refer to section 'Weaknesses'.

---

> ### Author Response · Authors · 2024-11-22
> **Responses to Reviewer hxch**
>
> We thank the reviewer for the careful reading and thoughtful comments. We address the reviewer's questions in the following. We hope the responses below address the reviewer's concerns.
>
> **Q1:** Only one baseline was considered, specifically "DP few-shot generation".
>
> **A1:** Thank you for the comment. Tang et al. (2023) represents the first work that generates synthetic few-shot demonstrations from a private dataset to perform ICL. We thoroughly compare our method to this baseline. In Section 2, we provide a literature review on Differentially Private ICL and find that they address different problems and cannot be compared with our algorithm, as listed below:
>
> **Duan et al. (2023):** This work *assumes the existence of an unlabeled public dataset*, which contrasts with our approach that requires no public data.
>
> **Wu et al. (2023):** This method *consumes the privacy budget per query*, limiting the number of queries that can be answered within a given budget. This makes it unsuitable for our focus on scenarios with an unlimited number of queries.
>
> **Hong et al. (2023):** This work proposes methods to generate *private and transferable instructions*, while our focus is on generating synthetic ICL examples.
>
> **Carey et al. (2024):** This study focuses on protecting *tabular data* used for ICL, while our focus is on text data.
>
> Therefore, we have focused on a detailed comparison with the available baseline. We would appreciate it if the reviewer could point us to some potential baselines that we are unaware of.
>
> **Q2:** If I am not mistaken, the iterative aspect is not utilized significantly, as $\hat{T}$ is frequently set to 1.
>
> **A2:** Thank you for the insightful comment. You are correct that in many cases $\hat{T}$ is set to 1. This is because increasing $\hat{T}$ introduces more iterations of private radius coverage check and projected mean estimation, which increases the overall privacy cost. To balance utility and privacy, we primarily set $\hat{T}\in\\{1, 2\\}$ in our experiments.
>
> That said, the iterative aspect is meaningful and can enhance performance in specific scenarios. For example, in Table 8 (TREC dataset, $\varepsilon=8$), the best result is achieved with $\hat{T}=2$. Similarly, in Table 9 (MIT-G dataset, $\varepsilon=8$), $\hat{T}=2$ also produces the best outcomes. These examples show that increasing $\hat{T}$ can improve performance when the privacy budget allows, highlighting the importance of the iterative approach in specific contexts. We appreciate the opportunity to clarify this aspect.
>
> ---
>
> We thank the reviewer again for the helpful comments and suggestions for our work. If our response resolves your concerns to a satisfactory level, we kindly request the reviewer to consider raising the rating of our work. Certainly, we are more than happy to address any further questions that you may have.

---

> > ### Comment · Reviewer_hxch · 2024-11-24
> >
> > **Thank you for the comment. Tang et al. (2023) represents the first work that generates synthetic few-shot demonstrations from a private dataset to perform ICL. We thoroughly compare our method to this baseline. In Section 2, we provide a literature review on Differentially Private ICL and find that they address different problems and cannot be compared with our algorithm, as listed below: ...**
> >
> > I would like to thank you for your response. However, Duan et al. (2023) also propose in the same paper an approach called 'PromptDPSGD,' which does not require access to public datasets. I would be interested to know why the authors did not consider this approach in their work.
> >
> > **Thank you for the insightful comment. You are correct that in many cases
> >  is set to 1. This is because increasing introduces more iterations of private radius coverage check and projected mean estimation, which increases the overall privacy cost. To balance utility and privacy, we primarily set  in our experiments.**
> >
> > Thank you for your response.

---

> > > ### Author Response · Authors · 2024-11-24
> > >
> > > Thank you for your question. We do not compare with PromptDPSGD (Duan et al., 2023) because it employs a **fundamentally different type of LLM prompting**. Specifically, PromptDPSGD focuses on **soft prompts**—trainable continuous embeddings—while our work uses **discrete prompts**, which are natural language instructions with task examples. Since current APIs [1,2,3,4] do not support soft prompting and only offer black-box access through discrete prompts, our approach is more practical for existing API-based LLMs.
> > >
> > > Thank you again for your feedback, and we are happy to address any further questions you might have.
> > >
> > >
> > > [1] OpenAI. Gpt-4 technical report, 2023.
> > >
> > > [2] Google. Lamda: Towards safe, grounded, and high-quality dialog models for everything. Website, 2023.
> > >
> > > [3] Antropic. Introducing claude. Antropic Website, 2023.
> > >
> > > [4] Brown, Tom B., et al. "Language models are few-shot learners." Proceedings of the 34th International Conference on Neural Information Processing Systems. 2020.

---

> > > > ### Author Response · Authors · 2024-11-25
> > > > **A gentle reminder**
> > > >
> > > > Dear Reviewer hxch,
> > > >
> > > > We've taken your feedback into careful consideration in our response. Could you please check whether our response has properly addressed your concern? If so, could you please kindly consider increasing your initial score accordingly? Certainly, we are more than happy to answer your further questions.
> > > >
> > > > Thank you for your time and effort in reviewing our work!
> > > >
> > > > Best Regards,
> > > >
> > > > Authors

---

> > > > > ### Comment · Reviewer_hxch · 2024-11-27
> > > > >
> > > > > Thank you for addressing my concerns in the rebuttal. Everything is now well-resolved, and I’ve updated my score accordingly!

---

> > > > > > ### Author Response · Authors · 2024-11-27
> > > > > > **Thank you!**
> > > > > >
> > > > > > We thank the reviewer very much for checking our responses and kindly increasing the rating. Many thanks for your effort into the review process.

---

### Official Review · Reviewer_Kogo · 2024-11-05

**Soundness:** 4
**Presentation:** 4
**Contribution:** 3
**Rating:** 8
**Confidence:** 5

**Summary:**

In this work, the authors focus on the privacy problem involved in potential leakage of in-context learning (ICL) examples for LLMs. Building on the prior work in the literature, the authors introduce a novel data-adaptive differentially private (DP) algorithm that can generate synthetic samples from the private dataset to be used in the ICL framework. Their extensive empirical studies demonstrate that this novel approach significantly improves prior work and also gets very close to the non-private ICL baselines.

**Strengths:**

The paper is well organized and focus on an important problem in privacy where jailbreaks attempts on LLMs can lead to the leakage of prompt which can lead to the leakage of private data samples presented in the prompt as few-shot manner.

Their approach is a novel improvement of prior work by making the synthetic generation of tokens "data-adaptive" as opposed to applying the same amount of noise as in the prior work.

This approach results in significant improvements in the private ICL performance as shown in a wide range of empirical studies and also gets close to the non-private ICL baselines.

**Weaknesses:**

I don't have too much to say here. I think it'd be informative to investigate the trade-offs between directly DP fine-tuning and generating synthetic ICL samples with DP but I think that's beyond the scope of this paper.

**Questions:**

Is there any good intuition on how to best divide the overall privacy budget across the privacy-consuming steps of Algorithm 1?

---

> ### Author Response · Authors · 2024-11-22
> **Responses to Reviewer Kogo**
>
> We thank the reviewer for the careful reading and thoughtful comments. We address the reviewer's questions in the following and have revised the paper accordingly. The changes are marked in blue in our revision. We hope the responses below address the reviewer's concerns.
>
> **Q1:** I think it'd be informative to investigate the trade-offs between directly DP fine-tuning and generating synthetic ICL samples with DP but I think that's beyond the scope of this paper.
>
> **A1:** We thank the reviewer for the insightful comment on the trade-offs between direct DP fine-tuning and generating DP synthetic ICL samples. The choice between these approaches depends on several factors, such as the degree of access to the LLM, computational resources, and the nature of the downstream task. For example, direct DP fine-tuning may be infeasible when only API access to the LLM is available, as model weights are not accessible. Additionally, generating DP synthetic ICL samples offers a lightweight and efficient solution, reducing the heavy computational burden associated with fine-tuning. However, certain downstream tasks might benefit more from fine-tuning, as it allows for stronger adaptation to specific domains. We appreciate the opportunity to discuss this and will consider a deeper exploration of these trade-offs in future work.
>
> **Q2:** Is there any good intuition on how to best divide the overall privacy budget across the privacy-consuming steps of Algorithm 1?
>
> **A2:** Thank you for the question. We allocate the overall privacy budget across three key steps of Algorithm 1: (1) GoodRadius in Line 5 with noise multiplier $\sigma_0$, (2) projected mean estimation in Line 20 with noise multiplier $\sigma_1$, and (3) radius coverage check in Line 12 with noise multiplier $\sigma_2$. To ensure the required $(\varepsilon,\delta)$-DP guarantee, we first set $\sigma_0$ and $\sigma_2$, and then compute $\sigma_1$ accordingly. Generally speaking, we set $\sigma_0$ and $\sigma_2$ to be relatively large compared with $\sigma_1$, to allocate more privacy budget to the projected mean estimation step. Below, we provide intuition on how the privacy budget is allocated.
>
> The GoodRadius step initializes a reference radius for the projected ball. Since it is not the final radius, this step can tolerate higher noise levels. Therefore, we typically set $\sigma_0$ to a large value (e.g., 10) to save privacy budget for the more sensitive projected mean estimation step.
>
> The radius coverage check verifies whether a sufficient number of the original probability vectors $p_\text{priv}^1,\ldots,p_\text{priv}^M$ lie within a small radius from $\tilde{p}_\text{priv}$. This involves counting the number of covered vectors, adding Gaussian noise with standard deviation $\sigma_2$, and comparing the noisy count to the total number of vectors $M$. When $M$ is large, the relative impact of the noise added to the count diminishes because the noise becomes negligible compared to $M$. Therefore, as $M$ increases, we can set $\sigma_2$ to a larger value. For example, when $M=40$, $\sigma_2$ can be set to around 5 without significant loss of accuracy.
>
> In conclusion, our privacy budget allocation prioritizes the projected mean estimation step for high accuracy, while allowing other steps to tolerate higher noise.
>
> We thank the reviewer for the insightful comment and we have added this discussion to our updated draft (Lines 918–931).
>
> --------------
>
> We thank the reviewer again for the helpful comments for our work. We are more than happy to address any further questions that you may have.

---

> > ### Comment · Reviewer_Kogo · 2024-11-25
> > **Response to the authors**
> >
> > Thank you very much for your insightful comments.

---

> > > ### Author Response · Authors · 2024-11-25
> > > **Thank you!**
> > >
> > > Thank you for your strong support of our paper! Your comments have greatly helped us improve the quality of our work.

---

### Meta-Review · Area_Chair_xd5P · 2024-12-21

**Metareview:**

This submission proposes a data-adaptive differentially private algorithm that is used to generate synthetic examples from a private dataset, which are then used to perform in-context learning (ICL).
Overall, the paper studies an important and natural problem, and proposes a novel algorithm that provides a clear improvement over the state-of-the-art. While the paper had some weaknesses (see the "Additional Comments On Reviewer Discussion" section below), they were satisfactorily addressed by the authors during the rebuttal phase.

**Additional Comments On Reviewer Discussion:**

Some concerns were raised by the reviewers including:
1) the submission’s contribution being incremental
2) the limited baselines in the empirical evaluation
3) a confusing experimental setup
4) the additional hyperparameters compare prior work

The authors have clarified most of these during the discussion, and there is agreement among the reviewers that the pros of the paper outweigh its cons.

---

### Decision · Program_Chairs · 2025-01-22

Accept (Poster)